# Vesicular glutamate release from central axons contributes to myelin damage

Sean Doyle[1], Daniel Bloch Hansen[1], Jasmine Vella[2], Peter Bond[1], Glenn Harper[1], Christian Zammit[2], Mario Valentino[2] & Robert Fern [1]

The axon myelin sheath is prone to injury associated with $N$-methyl-D-aspartate (NMDA)-type glutamate receptor activation but the source of glutamate in this context is unknown. Myelin damage results in permanent action potential loss and severe functional deficit in the white matter of the CNS, for example in ischemic stroke. Here, we show that in rats and mice, ischemic conditions trigger activation of myelinic NMDA receptors incorporating GluN2C/D subunits following release of axonal vesicular glutamate into the peri-axonal space under the myelin sheath. Glial sources of glutamate such as reverse transport did not contribute significantly to this phenomenon. We demonstrate selective myelin uptake and retention of a GluN2C/D NMDA receptor negative allosteric modulator that shields myelin from ischemic injury. The findings potentially support a rational approach toward a low-impact prophylactic therapy to protect patients at risk of stroke and other forms of excitotoxic injury.

[1] University of Plymouth, Plymouth PL6 8BY, UK. [2] University of Malta, Msida MSF 2080, Malta. Correspondence and requests for materials should be addressed to R.F. (email: Robert.fern@plymouth.ac.uk)

Myelin is an insulating, low-capacitance layer that wraps around the axon cylinder and is essential for fast action potential conduction. Myelin injury is fundamental to the functional loss the white matter of the CNS experiences in multiple sclerosis, trauma, and stroke[1–3]. Surprisingly, N-methyl-D-aspartate (NMDA)-type glutamate receptor (GluR) expression levels in myelin are comparable to those at neuronal synapses[4–6], are found in humans[6], and confer an elevated injury sensitivity under conditions of high extracellular glutamate[4,5,7,8]. The established view is that the glutamate release responsible for over-activation of myelinic NMDA GluRs under pathological conditions occurs via reverse glutamate uptake[9–11]. However, extracellular glutamate concentration has not previously been directly recorded within white matter while recent reports highlight vesicular release as an alternative potential source of white matter glutamate[12–14]. Under ischemic conditions, axonal vesicular glutamate release will empty directly onto the myelin sheath and may be trapped within the peri-axonal space between the axolemma and the internal myelin surface. The spatial characteristics of axon vesicular glutamate release may therefore be particularly relevant to myelin pathology. Myelinic NMDA receptors incorporate GluN2 C and D subunits and should be sensitive to selective negative allosteric modulators. GluN2C/D-containing NMDA receptors are generally extra-synaptic while negative allosteric modulators exhibit use-dependent properties. These features suggest that a putative myelinic NMDA receptor/axo-vesicular glutamate pathway may be particularly amenable to therapeutic intervention with clinical potential.

## Results

### Vesicular glutamate release in white matter is primarily axonal.

Vesicular docking in white matter was examined via live two-photon confocal imaging of fluorescent FM4-64[15] in corpus callosum axons of adult Thy-1/YFP mice. Low magnification images revealed extensive axonal FM4-64 loading, with individual axons reliably imaged at higher magnification (Fig. 1a–d). A similar approach using GFAP-GFP mice revealed fibrous white matter astrocytes with low levels of FM4-64 staining compared to neighboring axons (Fig. 1e, f; Supplementary Fig. 1d, e). FM4-64 de-staining (vesicular docking) was evoked by 50 mMK$^+$ perfusion, producing rapid docking in corpus callosum axons (significant within 60 s), but not in astrocytes (Fig. 1g). The [K$^+$] used to evoke this response is comparable to that recorded during brain ischemia[16], and produced axon depolarization sufficient to reversibly block excitability (Supplementary Fig. 1a–c). To our knowledge, this is the first direct comparison of vesicular fusion within different cellular components in white matter, and it reveals extensive fusion in axons compared to their companion astrocytes and is consistent with earlier observation of vesicular release from axons[15]. We have shown previously that YFP(+) axons in this model are myelinated[17] and the diameter of YFP(+) axon in the current study had a range considerably higher than the upper limit for non-myelinated axons in mouse corpus callosum[18] (Supplementary Fig. 1f). As there was no evidence of localized de-staining along YFP(+) axons imaged in long-section (L-S), the majority of vesicular fusion in these axons must occur under the myelin sheath and empty into the periaxonal space. This mechanism was also triggered by ischemic conditions where significant axonal release was found after 15 min (Fig. 1h).

The glutamate concentration in the extracellular space ([glutamate]$_e$) increased by $13.0 \pm 3.6\,\mu M$ during perfusion with 50 mM [K$^+$] (Fig. 1i–k), a rise that was not inhibited by glutamate transport inhibition (200 $\mu$M TBOA) but was significantly reduced by inhibition of vesicular glutamate loading (50 nM bafilomycin; Fig. 1i–k). Note, axonal glutamate transporters are localized primarily to the node of Ranvier rather than the internodal region[19] and TBOA is likely to access axonal transport sites relatively quickly. In the adult corpus callosum, TBOA evoked a $1.3 \pm 0.4\,\mu M$ peak increase in resting [glutamate]$_e$ indicating ongoing glutamate regulation in white matter via glutamate transport under physiological conditions (Supplementary Fig. 2).

FM4-64 axon imaging reported uniform vesicle fusion along axons with no focal sites of fusion that might indicate local glutamate release at nodes of Ranvier; vesicular glutamate release must therefore empty largely into the periaxonal space under the myelin sheath which covers 99% of the axon cylinder. Consistent with this, an established DiOC$_6$(3)/X-rhod-1 confocal imaging protocol[8] reported elevated [Ca$^{2+}$]$_i$ in the cytoplasmic compartment of the myelin sheath following depolarization with 50 mMK$^+$ (Fig. 2a, b). Myelin X-rhod-1 loading is largely peri-axonal[5,8] and the myelinic K$^+$-evoked [Ca$^{2+}$]$_i$ rise was prevented by pre-incubation with bafilomycin, indicating myelin calcium influx following vesicular glutamate release from axons. Ultrastructural analysis of long-section (L-S) rodent optic nerve (RON) axons revealed regions of 20–50 nm axoplasmic vesicle clusters within the internodal zones (Fig. 2c arrows); vesicles were not clustered within nodal regions and were not observed in glial processes aligned adjacent to myelinated axons. RONs exposed to 30 min of oxygen–glucose deprivation (OGD) prior to fixation contained significantly fewer internodal axoplasmic vesicles (Fig. 2d, f, g), and in these nerves the vesicles were found at the sub-myelinic axolemma occasionally caught in the act of membrane docking/fusion (Fig. 2d, e). As in control nerves, such vesicles were absent from glial processes neighboring the myelin (e.g., Fig. 2d). The ultrastructural and X-rhod-1 imaging data confirm internodal axoplasmic vesicle-axolemma docking, which will release glutamate into the periaxonal space beneath the myelin sheath and may lead to early focal myelin damage. Axon cylinders were largely unaffected after 30 min of OGD and retained normal microtubules, a feature of healthy axons (Fig. 2d, white arrows). Early signs of myelin damage were evident at sites where axoplasmic vesicles were present and included localized splitting and bubbling of the lamina (Fig. 2d, e*).

### White matter ischemic glutamate release is primarily vesicular.

The corpus callosum sits adjacent to gray mater structures such as the cortex (Fig. 1a) that have extensive glutamatergic input. To avoid the potential for spillover from gray matter synapses, we examined [glutamate]$_e$ in isolated RON, a white matter structure with no neuronal synapses or neighboring gray matter. Resting [glutamate]$_e$ was $8.9 \pm 1.4\,\mu M$ in adult rat RON, higher than the 1.3–4.5 $\mu$M range recorded in other white matter preparations (Supplementary Fig. 2a). Glutamate biosensor electrodes were found to be anoxia-sensitive (Supplementary Fig. 3) and ischemia was therefore modeled using combined aglycemia/oxidative phosphorylation block, producing a 22.4–27.5 $\mu$M glutamate rise (Fig. 3a, b). Block of glutamate transport (TBOA or zero-Na$^+$) did not significantly reduce the ischemic glutamate increase, but release was both Ca$^{2+}$- and voltage-gated Ca$^{2+}$ channel (100 $\mu$M diltiazem)-dependent as predicted for vesicular release (Fig. 3a, b). Recording from adult RON required a long period of stabilization following sensor insertion (Supplementary Fig. 4c), and ischemic [glutamate]$_e$ was more extensively examined in juvenile rat RON, which stabilized more quickly. At this age, ischemia evoked a $5.6 \pm 0.8\,\mu M$ [glutamate]$_e$ rise that was not significantly affected by block of glutamate transport (TBOA, or zero-Na$^+$), swelling-mediated glutamate release (5 mM furosemide), cysteine-glutamate antiport (250 $\mu$M SAS), P2X7-, pannexin- and connexin-channels (100 $\mu$M CBX), or swelling-operated channels

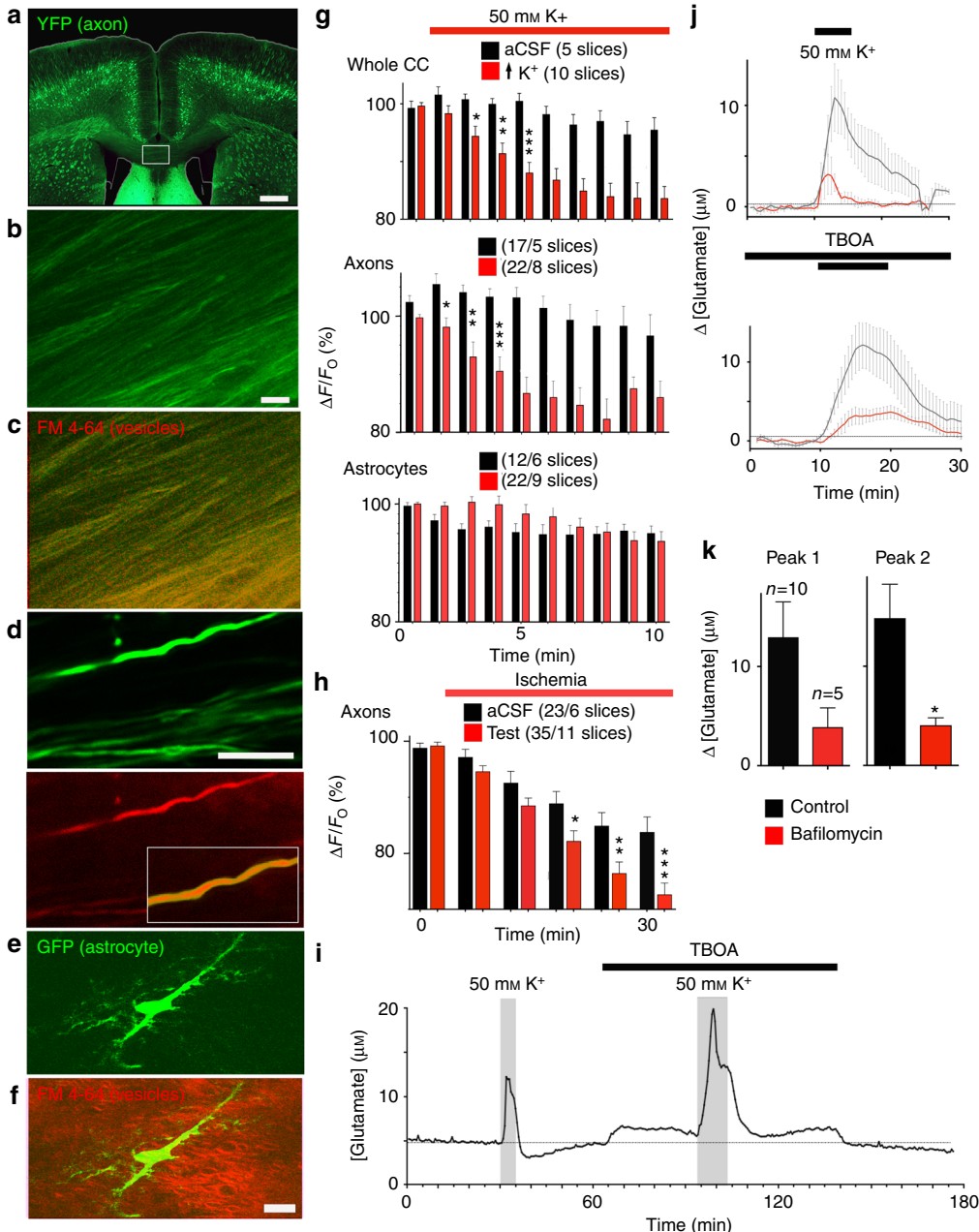

**Fig. 1** Axons are the principle site of white matter vesicular fusion. **a** Thy-1-YFP expression in adult coronal section (box: corpus callosum). **b**, **c** YFP(+) corpus callosum axons (**b**) and FM4-64 vesicles (**c** overlaid red). **d** Single axons (overlaid at higher gain in the box). **e**, **f** Corpus callosum astrocyte (**e**) in adult GFAP-GFP mouse and FM4-64 staining (**f** overlaid red). **g** K$^+$-evoked FM4-64 de-staining in whole corpus callosum, axons, and astrocytes. **h** Corpus callosum axon FM4-64 de-staining in aCSF and during ischemia. **i** K$^+$-evoked [glutamate]$_e$ release is not inhibited by TBOA. **j** Both the first (top) and second (bottom) 50 mMK$^+$-evoked glutamate rise is reduced by bafilomycin (50 nM, red trace). **k** Data summary shown on the bottom. Scale = 1 mm (**a**), 10 μm (**b**, **d**, **f**). Stars indicate the first mean to reach significance. Grouped analysis ANOVA; correct for multiple comparisons using the Holm−Šídák method. P values: **g** top *0.016, **0.001, ***0.0000; **g** middle *0.038, **0.002, ***0.0004; **h** *0.021, **0.009, ***0.0013. Unpaired t-test P values: **k** Peak 1 ns 0.1119; Peak 2 *0.048

(100 μM NPPB) (Fig. 3c, d). Glutamate release was Ca$^{2+}$-dependent and was significantly reduced by blockers of vesicular glutamate release (50 nM bafilomycin or 500 nM rose bengal; Fig. 3e, f), confirming the significance of vesicular release to ischemic [glutamate]$_e$ elevation in white matter.

**NMDA receptors mediate acute myelin injury during ischemia.** Drug penetration into the peri-axonal space is known to be slow. For example, diffusion from the node of Ranvier into the peri-

nodal space that is contiguous with the periaxonal space takes several hours[20]. Drugs may also access the peri-axonal space directly through the myelin sheath if they are lipid soluble, e.g. the NMDA blocker MK-801 that associates/dissociates with synaptic lipid membrane with a time constant ($\tau$) of ~4 min[21], indicating $\tau = 56$ min for penetration through the sheath of a typical mature RON axon. Prior studies have shown that short periods of NMDA-receptor block fail to protect adult white matter from ischemic injury[11,22], and we also found no protection following a 20 min pre-treatment period with MK-801 (Fig. 4b). However,

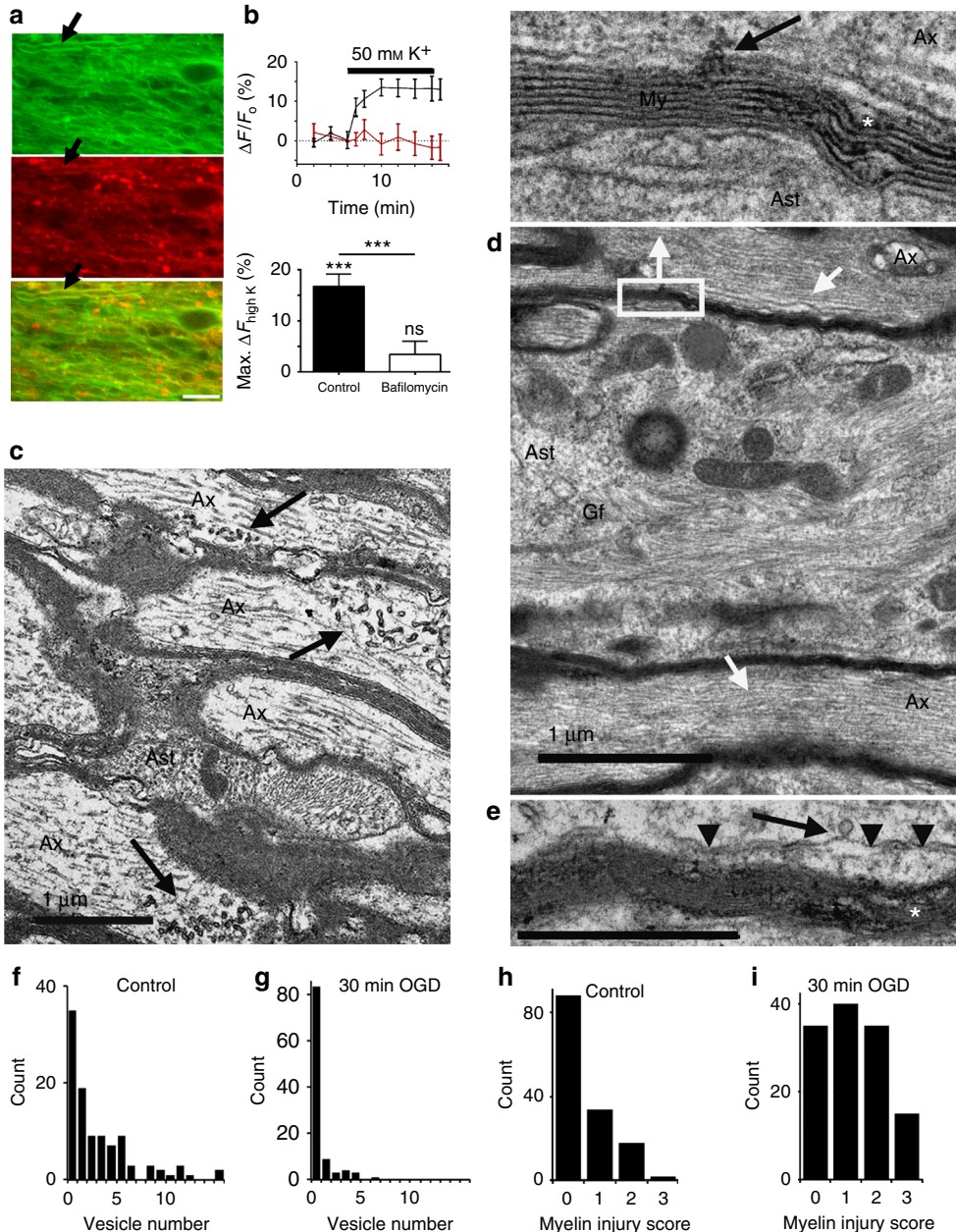

**Fig. 2** Vesicular release under the myelin. **a** Myelin loaded with $DiOC_6$ (green) and the $Ca^{2+}$-indicator X-rhod-1 (red) in adult mouse RON. Note the loading into myelin profiles (e.g., arrow). **b** Top: X-rhod-1 intensity (relative to initial mean) is elevated following depolarization with $50\,mMK^+$ (black line) consistent with activation of NMDA receptors in the myelin, an effect blocked by pre-treatment with the vesicular loading blocker bafilomycin (red line). **b** Bottom: data summary. Asterisks on the error bars indicate $P = 0.0000$ significance vs., the initial fluorescence mean; on the bar they indicate $P = 0.0000$ significance vs., the two conditions. $n = 5$ mice in each protocol, 1–3 slices/mouse, ANOVA with Holm−Šídák post test. Bar $= 8\,\mu m$. **c** 20–50 nm vesicles are present in clusters in the sub-myelinic axoplasm (e.g., arrows) (Ax = axons; Ast = astrocyte process containing glial filaments; bar $= 1\,\mu m$). **d** Sub-myelinic vesicles are less common in RONs fixed after 30 min of OGD and can be seen juxtaposed on the axolemma (black arrow). White boxed area shown at higher magnification above. Note the absence of vesicles in the astrocyte process identified by the presence of glial filaments (Gf), and the retention of microtubules in myelinated axons (white arrows). Bar $= 1\,\mu m$. **e** A single vesicle (black arrow) docked with the axolemma (arrow heads) beneath the myelin following OGD. Note the presence of focal myelin injuries in both (**d**) and (**e**) (white asterisks). Bar $= 500\,nm$. **f, g** The distribution of vesicle clusters in axon profiles by number of vesicles in control and 30 min OGD RONs (per field of view). Note the large number of axons with zero vesicles following OGD (scales differ). **h, i** Focal myelin injury scores are shifted to higher values in myelin regions adjacent to axoplasmic vesicles post OGD compared to control RONs

MK-801 protection increased with pre-treatment time with maximal protection of >400% after 120 min pre-treatment (Fig. 4a, b). Myelinic NMDA GluRs contain the GluN2C/D subunit[4,5,7] and we tested 120 min pre-treatment with the GluN2C/D-specific antagonists PPDA (50 µM) and QNZ-46 (50

µM)[23], which also significantly increased recovery from ischemia (Fig. 4c). TRPA1 receptor block (10 µM A967076) has recently been shown to be protective against ischemic injury in oligo-dendrocyte processes and myelin sheaths[24], but did not provide functional white matter protection in our model of acute ischemic

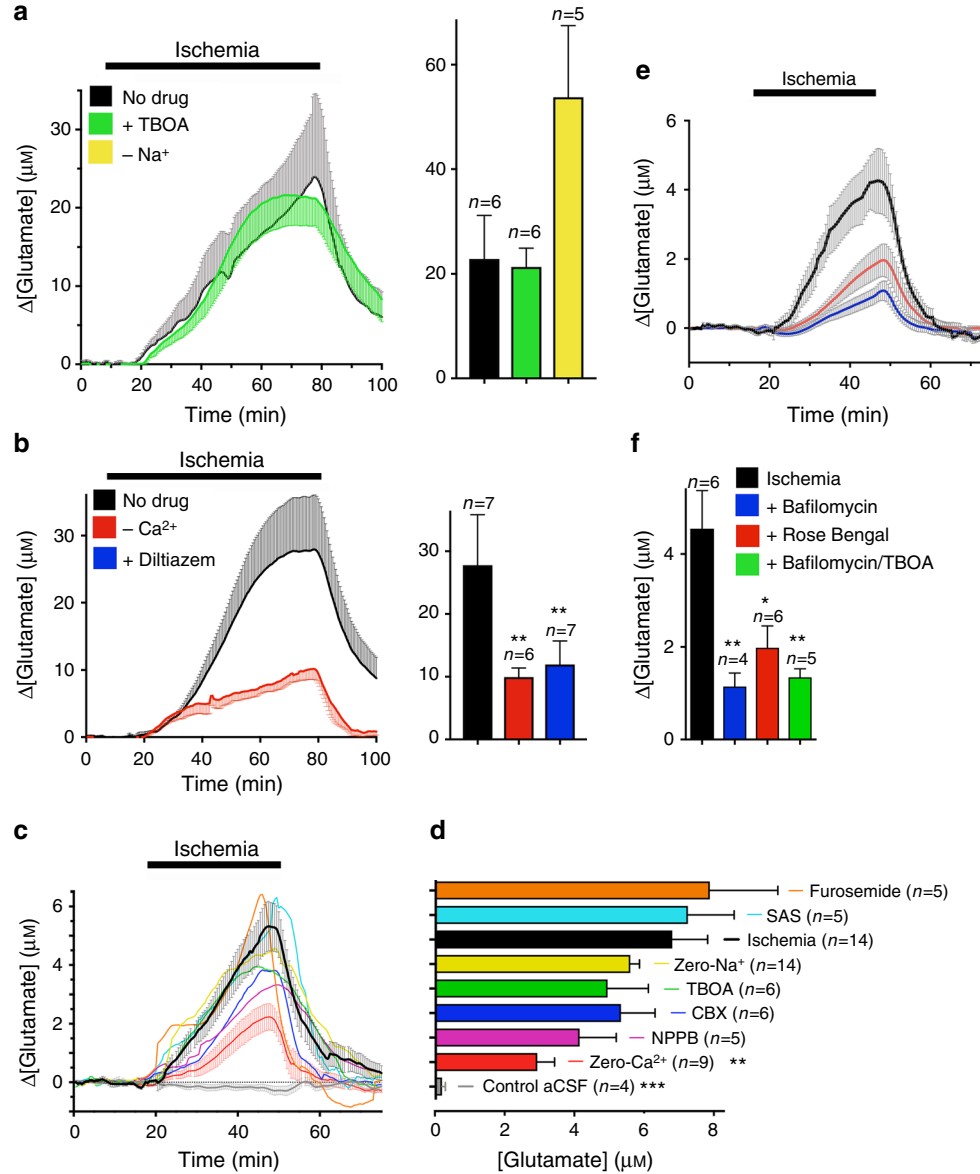

**Fig. 3** White matter ischemic glutamate release is primarily vesicular. **a** Ischemia-evoked $[\text{glutamate}]_e$ elevation in adult rat RON is $Na^+$-independent/ TBOA-insensitive. **b** The ischemic-$[\text{glutamate}]_e$ rise is $Ca^{2+}$-dependent (ANOVA with Holm−Šídák post test: $P = 0.012$) and diltiazem-sensitive ($P = 0.010$). **c, d** In juvenile rat RON, the ischemic $[\text{glutamate}]_e$ elevation ($P = 0.001$ vs., control aCSF) is $Ca^{2+}$-dependent ($P = 0.009$ vs., ischemia)/Na $^+$-independent and resistant to inhibitors of non-vesicular glutamate release pathways. **e, f** Ischemic-$[\text{glutamate}]_e$ elevation in juvenile RON is significantly inhibited by blockers of vesicular glutamate loading bafilomycin ($P = 0.004$); rose bengal ($P = 0.013$) and combined bafilomycin + TBOA ($P = 0.004$)

injury (Fig. 4c). Previous studies of oligodendrocyte injury have often relied on live imaging of the cells cytoplasmic domain[5,7,24]. Simultaneous live imaging of the myelinic and cytoplasmic domains of oligodendrocytes revealed that they are largely distinct (Fig. 4d) and that morphological and structural changes in oligodendrocyte processes imaged via fluorescence label expression do not reflect myelin pathology. Dual domain imaging during ischemia showed the myelin decompaction predicted by prior ultrastructural studies (Fig. 4e, f), with ongoing disruption of cell processes/somata swelling. Myelin thickness increased from 0.67 µm ±0.02 to 0.90 µm ±0.07 after 60 min ischemia +60 min recovery, an effect that was prevented by 120 min pre-treatment with QNZ-46 (Fig. 4e, f). Ultrastructural analysis revealed extensive myelin decompaction and bubbling following the standard OGD protocol, in addition to disruption of glial

processes and soma (Fig. 5a–c). Bubbling of the inner myelin layer was particularly evident following OGD (Fig. 5c) and involved regions where the inner myelin layer detached from the remaining compact myelin and formed a series of bubbles. Myelin bubbling was accompanied by a reduction in the number of myelin layers (Fig. 5c small arrows). Axon cylinders in these regions (Fig. 3c Ax) often retained microtubule profiles and a clearly contiguous axolemma separating the cylinder from the myelin bubbles. The myelin protection recently reported following TRPA1 block did not extend to structures defined as axo-plasmic vesicles[24], which appear to have similar features to the myelin bubbles we here identify under the remaining compact myelin sheath. 120 min pre-treatment with QNZ-46 almost entirely prevented these structural changes (Fig. 5d–f). Quanti-tative analysis of myelin decompaction via G-ratio analysis of

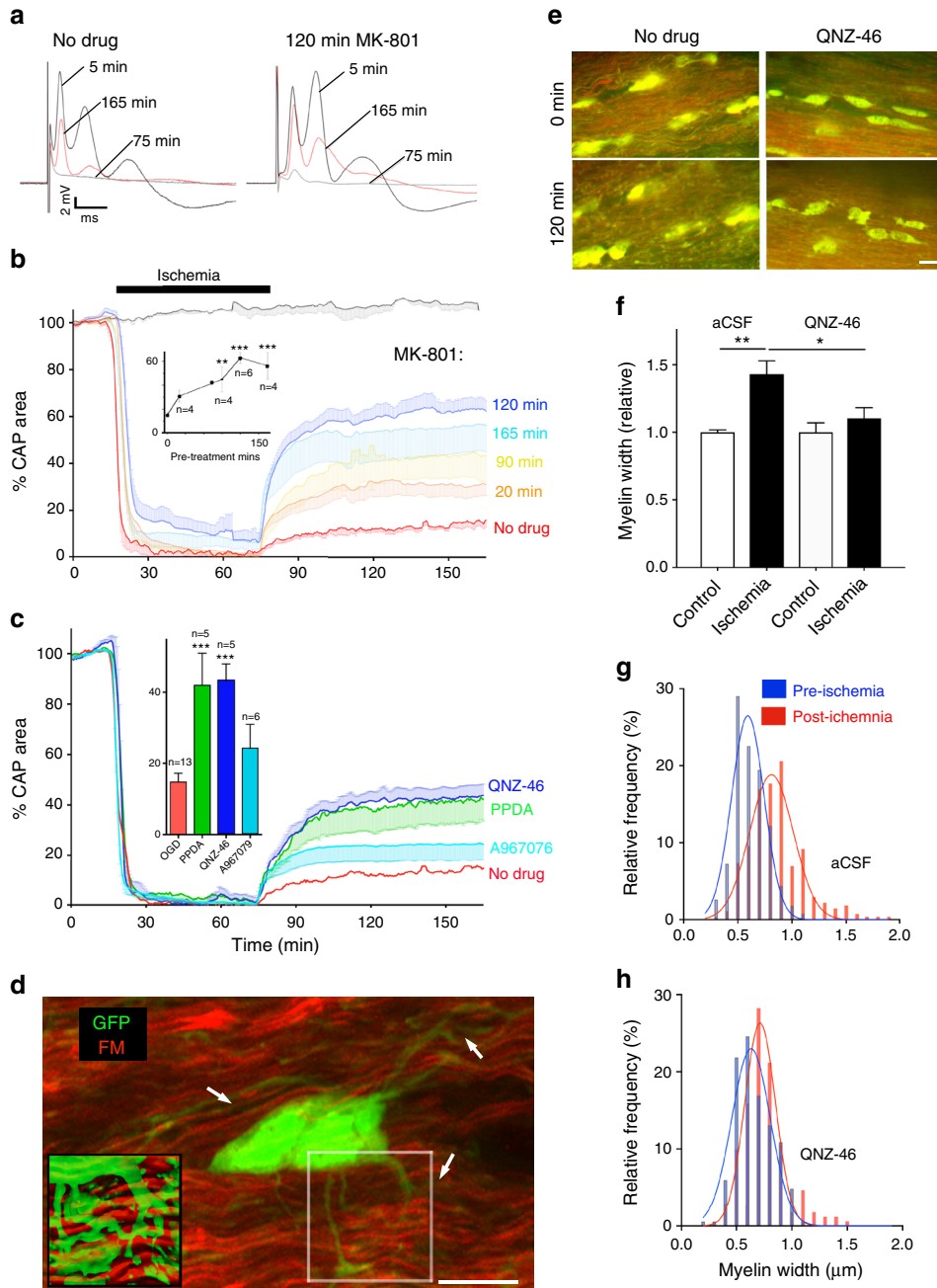

**Fig. 4** Myelin injury is NMDA-receptor-mediated. **a** CAP recordings from adult rat RON showing elevated functional recovery from ischemia following 120 min MK-801 pre-treatment. **b** Ischemia-evoked loss and recovery of function following different periods of MK-801 pre-treatment (insert: data summary). ANOVA with Holm−Šídák post test. P values: 90 min **0.002; 120 min ***0.0001; 165 min ***0.0000. **c** 120 min pre-treatment with selective NMDA receptor GluN2C/D blockers QNZ-46 (P = 0.0013) or PPDA (P = 0.0009), or the TRPA1 blocker A967076. **d** Oligodendrocyte cytoplasmic domain (PLP-GFP mouse, arrows) is distinct from the myelin domain (FM: fluoromyelin red) in live adult mouse RON. Highlighted square shown in insert as rendered stack. **e** Myelin structure is protected from ischemic damage by 120 min pre-treatment with QNZ-46, imaged via FM vital staining. **f**–**h** Ischemia evoked myelin swelling (P = 0.000 in aCSF) is prevented by 120 min QNZ-46 pre-treatment (P = 0.047 vs. aCSF). Scale = 10 μm

axon cross-section (X-S) confirmed the myelin protecting properties of the drug (Fig. 5g–k). GluR-mediated myelin loss during OGD was confirmed in the RON by diminished intensity of the established myelin stain FluoroMyelin Red (FM) (Fig. 5l).

QNZ-46 is a 4-oxo-3(4H)quinazolinyl derivative containing the trans-stilbene pharmacophore[23] and has the lipophilicity common to fluorescent myelin stains[25], although with reduced aromaticity in the A ring (Supplementary Fig. 5a). The structure

also contains the quinazolinone backbone known to exhibit strong fluorescence[26] and the drug therefore has the structural components of a fluorescent myelin stain. QNZ-46 had a peak emission at 450 nm in a lipid environment (Supplementary Fig. 5b), allowing drug uptake to be monitored in real time. QNZ-46 loaded into adult rat RON from the bath over 120 min and was retained following wash-out (Fig. 6a). After 120 min of bath loading into brain slices, vital QNZ-46 fluorescence was localized

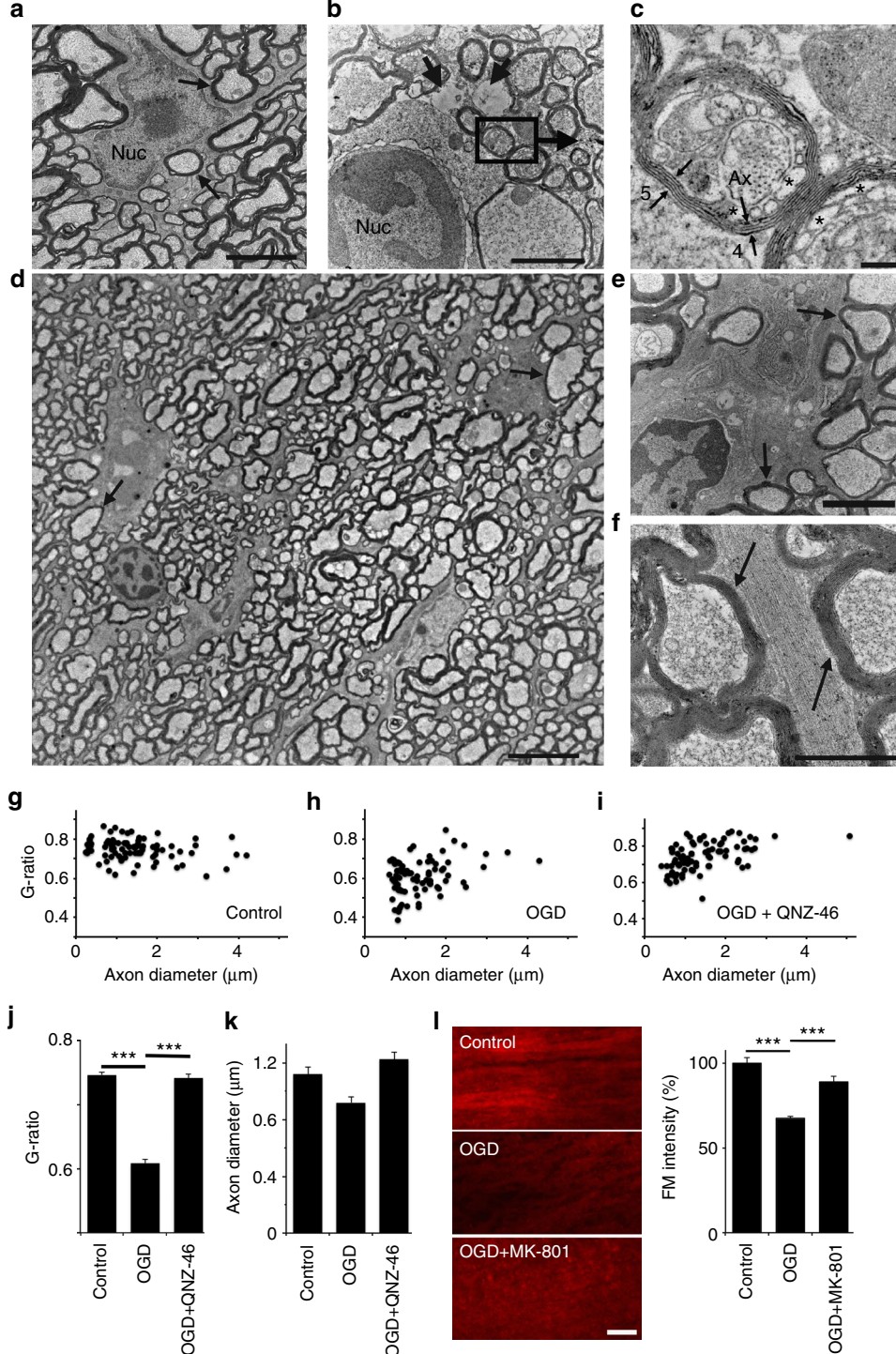

**Fig. 5** Myelin protection in adult mouse RON. **a** Oligodendrocyte (Nuc = nucleus) and myelinated axons (e.g., arrows) in control RON have a normal healthy structure. **b**, **c** Following OGD there is glial disruption including nuclear condensation and process degeneration (e.g., short arrows) with myelin disruption in the axon population. Note that the inner myelin layers are often separated and may form bubbles. For example in (**c**) (higher power image of the boxed area in (**b**)) where a region of myelin on the left side of the axon has five layers (small arrows) and no bubbles but the inner layer has four layers with a series of bubble profiles on the right side (e.g., *). Note retention of microtubules within the axon profile (Ax). **d–f** Uniform protection of myelin structure (e.g., arrows) throughout RONs fixed after OGD + QNZ-46. Note that glial cell soma (**d**, **e**) and processes (**f**) also retain normal structure in this protocol. **g–j** Myelin thickness assessed as G-ratio under the three conditions, showing myelin expansion following OGD (ANOVA with Holm−Šídák post test; $P = 0.0001$) and prevention of this effect by perfusion with QNZ-46 ($P = 0.0007$). **j**, **k** Mean data showing the changes in G-ratio under these conditions (**j**); which are not accounted for by significant axonal shrinking (**k**). **l** Myelin stained with fluoroMyelin (FM) under these conditions (left) and (right) the mean FM intensity decline evoked by OGD ($P = 0.00027$), which is prevented by MK801 ($P = 0.00062$). Bar **a**, **b**, **e**, **f** = 1 μm; **d** = 5 μm; **c**, **l** = 100 nm

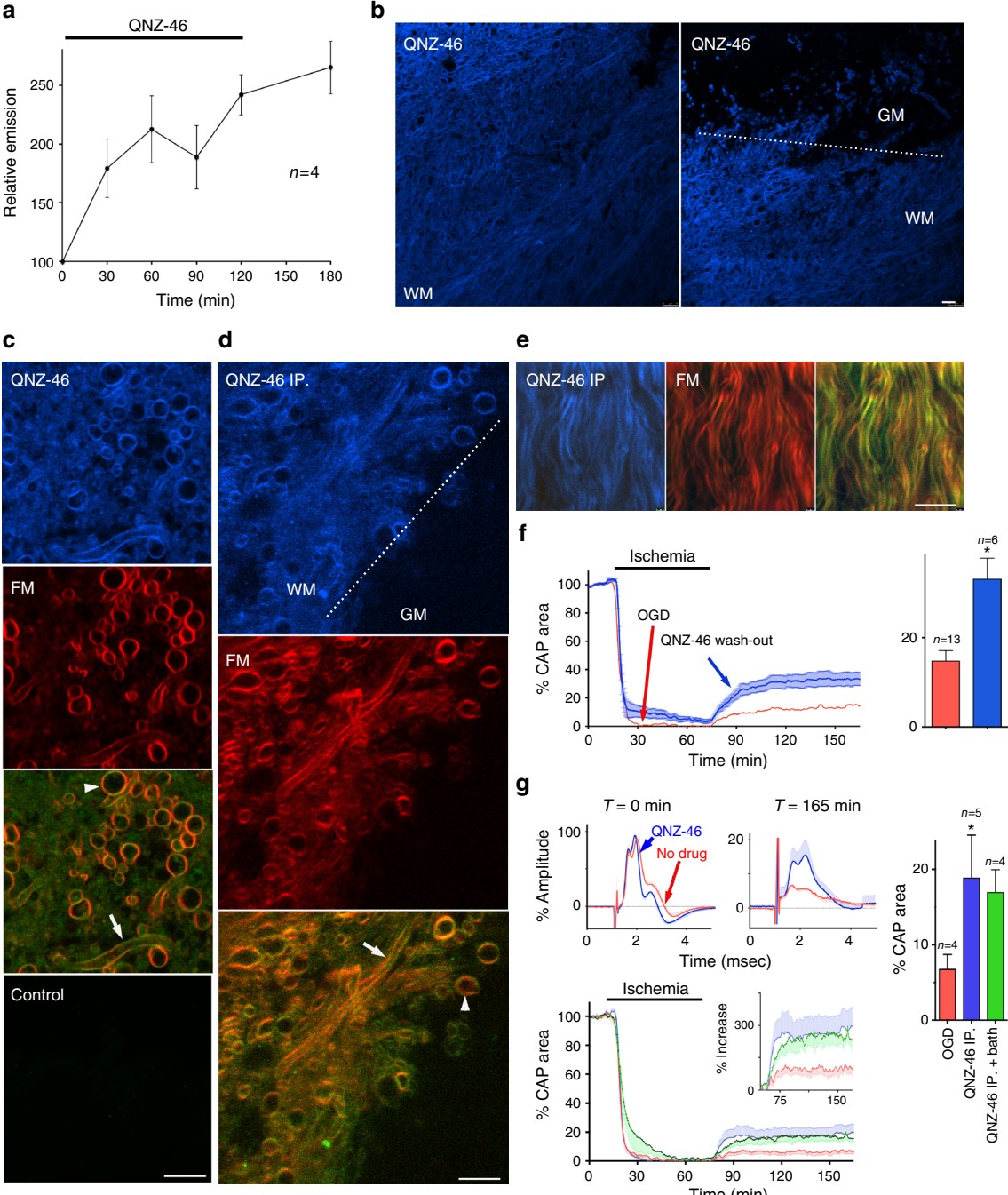

**Fig. 6** QNZ-46 is absorbed by myelin. **a** QNZ-46 emission shows accumulation and retention in adult rat RON. **b** Confocal imaging of vital QNZ-46 fluorescence (120 min treatment) in adult corpus callosum (left) and the white matter–gray matter border (right). **c** Vital QNZ-46/FM co-staining of myelinated axon profiles. **d**, **e** Vital QNZ-46/FM co-staining in brain slice (**d**) and RON (**e**) following systemic QNZ-46 injection 4 h pre-sacrifice. **f** 60 min QNZ-46 pre-treatment + 60 min wash-out is functionally protective of adult rat RON (ANOVA with Holm−Šídák post test; $P = 0.037$). **g** Systemic QNZ-46 pre-treatment is functionally protective of mouse RON perfused with aCSF (blue, $P = 0.047$), and is not significantly different when QNZ-46 is included in the bath (green). Note, proportional recovery was similar to that found in Fig. 3c (insert). Merged images recolored for clarity. Scale bars = 5 μm

to myelin axon profiles, was low in gray matter regions, and co-localized with FM (Fig. 6b, c). Following i.p. injection (20 mg/kg in 50/50 DMSO/β-cyclodextrin 240 min prior to the killing, based on known CNS action of a similar compound[27]), vital QNZ-46 fluorescence had a similar distribution in mouse brain slices to that produced by bath loading (Fig. 6d, e), demonstrating brain penetration and myelin retention (tissue was dissected into QNZ-46 free aCSF for imaging).

QNZ-46 is an allosteric modulator of GluN2C/D-containing NMDA GluRs with use-dependent features[23,28]. It is the most selective inhibitor of GluN2C/D-containing GluR currently identified with a >50-fold IC50 differential over GluN2A/B-containing receptors. In silico modeling indicates drugability, although with a high polar surface area (Supplementary Fig. 5c). The myelin partitioning and trapping evident in Fig. 6a–e suggests QNZ-46 may provide myelin protection following

removal from the extracellular space. Indeed, 60 min pre-treatment followed by 60 min of wash-out significantly elevated functional recovery in the adult rat RON (Fig. 6f). No similar effect was found with the non-NMDA glutamate receptor blocker NBQX, an anti-excitotoxic drug thought to act at the

oligodendrocyte cell body[7] (Supplementary Fig. 6). Systemic i.p. injection of QNZ-46 followed by 240 min recovery, dissection and 60 min bath perfusion with aCSF also increased compound action potential (CAP) recovery in adult mouse RON from $6.8 \pm 1.9\%$ to $18.9 \pm 5.6\%$, representing a >270% increase compared to vehicle-

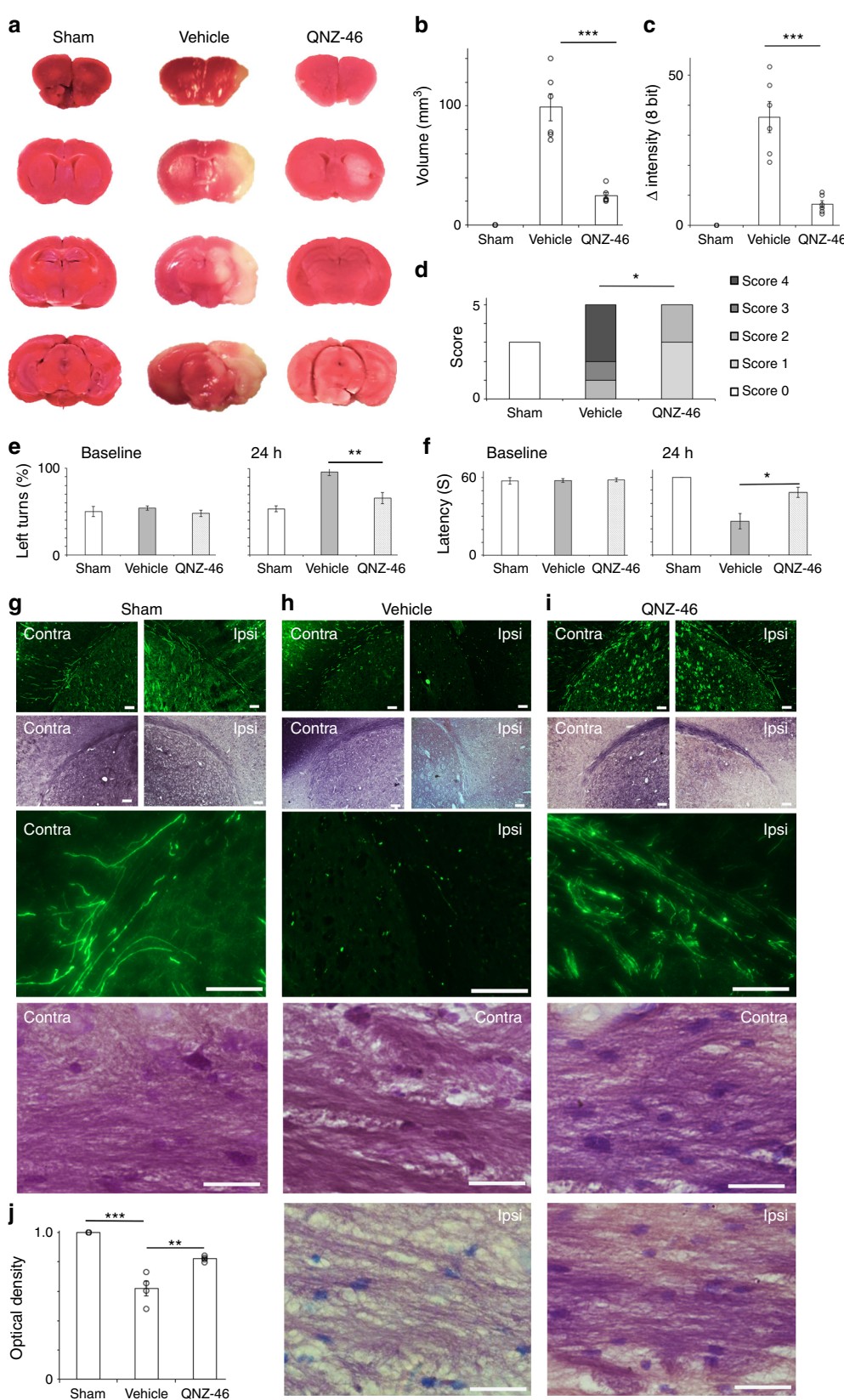

treated control (Fig. 6g). This effect was not potentiated by the presence of QNZ-46 in the bath, suggesting that systemic QNZ-46 pre-treatment and myelin trapping results in maximal protection (Fig. 6g). The efficacy of QNZ-46 trapped within myelin may suggest drug interaction with an NMDA receptor site within the lipid lamella/oligodendrocyte cell membrane. Alternatively, myelin may act as a reservoir that gradually releases QNZ-46 to act at extra- or intra-cellular sites on the receptor. In either case, our data suggest that incorporation of myelin-targeted elements such as trans-stilbene may enhance the effectiveness of myelin treatments generally.

Although crossing the blood−brain barrier and segregating into myelin, systemic pre-treatment with QNZ-46 produced no apparent behavioral effects or acute toxicity. A 120 min single dose pre-treatment protocol was tested in a standard 60 min transient middle cerebral artery occlusion (tMCAO) model of stroke. Brain lesions assessed 24 h post-reperfusion in vehicle treated wild-type mice included extensive damage to white matter structures such as the external capsule and gray matter regions such as the overlying motor cortex. Drug treatment greatly reduced lesion volume and improved the performance in behavioral tests compared to vehicle-treated controls (Fig. 7a±e). External capsule axon integrity assessed via YFP expression in Thy-1-YFP mice was lost within the lesion site in the vehicle cohort, but was largely preserved in the QNZ-46 treated cohort (Fig. 7g–j). Myelin integrity was protected to a similar extent (Fig. 7g–j).

A second series of tMCAO data was generated with mice perfusion fixed after the 24 h recovery for ultrastructural analysis. Lesion reduction following QNZ-46 pre-treatment was replicated in these experiments (lesion volume vehicle = $119.4 \pm 8.9$ mm$^3$, QNZ-46 = $38.9 \pm 7.0$ mm$^3$; $P < 0.0001$). Neuronal somata, neuropil, and white matter structure was well preserved in the external capsule-motor cortex border region of the contralateral hemisphere (Fig. 8c). Widespread cellular breakdown was apparent in the comparable vehicle-treated ipsilateral region (Fig. 8d, e), including necrotic cell death, neuropil degeneration, and universal myelin destruction in both white matter tracts and gray matter axons (Supplementary Fig. 7). In mice pre-treated with QNZ-46, all cellular elements were relatively preserved (Fig. 8f–h; Supplementary Fig. 7), with myelinated axons having no signs of myelin splitting or bubbling. Largely uninjured neuronal somata, neuropil, and glial somata were apparent throughout the region, although astrocyte processes (some containing glial filaments) were swollen and damaged (Fig. 8g, white arrows). Quantitative assessment of somata injury (which will include both neurons and glia since they cannot be reliably distinguished post vehicle-treated injury) showed QNZ-46 protection in the ipsilateral hemisphere of both white matter and gray matter areas, with no significant difference in somata damage between contra- and ipsilateral sides in white matter (Fig. 8i). Myelin expansion (G-ratio, not measured in ipsilateral white matter due to the extent of damage) was not present in the ipsilateral white matter of QNZ-46-treated mice. It is apparent from these in vivo experiments that a single dose pre-treatment with QNZ-46 protected both gray matter and white matter

structures to deliver a high level of structural and functional neuroprotection.

## Discussion

The results highlight the significance of vesicular glutamate release from axons and demonstrate the involvement of this phenomenon in ischemia-evoked myelin damage. Earlier reports have documented reverse glutamate release under ischemic conditions in the CA1 region of the neonatal hippocampus[9], adult spinal cord[10], and mouse optic nerve[11]. To our knowledge, the current report is the first to directly measure extracellular glutamate in white matter and while glutamate transport was found to be significant for homeostatic regulation of the neurotransmitter, we found no evidence for significant ischemic release via this mechanism. Ischemic glutamate release pathways in the white matter of the brain are likely to differ from those operating in gray matter areas such as the hippocampus CA1, while earlier white matter studies have generally examined secondary effects of reverse glutamate transport block and this may account for the discrepancy between earlier findings and the current results.

Approximately 95% of clinical strokes involve white matter, which accounts for ~49% of stroke total mean infarct volume[29]. Stroke in the territory of penetrating arteries preferentially target white matter and accounts for ~25% of stroke cases, representing the second leading cause of dementia[3,30]. White matter stroke features rapid myelin damage[31] and remyelination failure in white matter lesions significantly contributes to functional loss[3,30]. The mechanisms underlying myelin injury in these acute ischemic lesions have high clinical relevance and may share common features with other forms of myelin damage, for example those operating in multiple sclerosis and CNS trauma[32,33]. We have shown that acute ischemic myelin injury results from vesicular glutamate release from axons, leading to cytotoxic over-activation of GluN2C/D-containing myelinic NMDA GluRs preventable by the selective negative allosteric modulator QNZ-46 (Supplementary Fig. 8). QNZ-46 has high selectivity for GluN2C/D-containing GluRs, shows novel myelin accumulation and retention, is brain accessible, and has the basic features of a clinically useful drug, suggesting outstanding clinical potential against excitotoxic myelin injury. QNZ-46 exhibits persistent CNS protection, elevating injury tolerance after the drug is removed from the extracellular space, and has clinical potential for treatment of prevalent neurological disorders involving myelin damage in particular in patients at risk of ischemic injuries such as stroke.

While the neuroprotective effect of broad-spectrum NMDA GluR blockers is well established, these drugs have failed to translate into clinical practice. The reasons for this are complex and involve unacceptable side effects and the short therapeutic window that follows stroke onset. This second problem may be insurmountable; the FAST-MAG trial achieved paramedic delivery of the NMDA receptor blocker $Mg^{2+}$ within 45 min of stroke symptom onset but failed to improve outcomes[34]. Negative allosteric modulators such as QNZ-46 exhibit use-dependent block predisposing them to target pathological over-activation of

**Fig. 7** A single QNZ-46 pre-treatment produces high levels of white matter and gray matter neuro-protection. **a–c** Brain lesion volume 24 h post-tMCAO. **d–f** Functional recovery prior to the killing. Drug treatment significantly improved the outcome in all measures. ANOVA with Holm−Šídák post test. P values: **b** ***0.0000; **c** **0.000; **e** **0.004; **f** *0.011. Mann−Whitney test; **d** <0.05. **g–j** YFP expression (green) and luxol fast-blue/cresyl violet (blue/purple: myelin) in Thy-1-YFP mice. **g** YFP(+) axons project within the external capsule which is extensively myelinated in contralateral (Contra) and ipsilateral (Ipsi) hemispheres in sham-operated mice. Higher power micrographs are shown at the bottom. **h** Mice treated with vehicle show disruption and loss of axonal YFP that extends to both hemispheres and loss of myelin within the ipsilateral white matter. **i** Drug-treated mice retain YFP(+) axons and myelin staining in both hemispheres. **j** Myelin stain density is significantly reduced in the vehicle-treated group (***0.0002), and preserved by QNZ-46 pre-treatment (**0.005). Scale bars = 100 μm

receptors over normal physiological receptor function[23,24]. The physiological functions of oligodendrocyte NMDA GluRs includes regulation of GLUT-1 expression[22] and axo-glial myelin[35]. Constitutive targeted oblation of the obligatory Nr1 subunit in oligodendrocytes leads to a switch in the mechanism regulating myelination and a downregulation of GLUT-1 that is associated with gradual structural compromise; these effects may be avoided by this class of drug. In the adult CNS, GluN2C/D subunits are primarily incorporated into extra-synaptic NMDA receptors[36–38], and are expressed at lower levels in gray matter regions than are

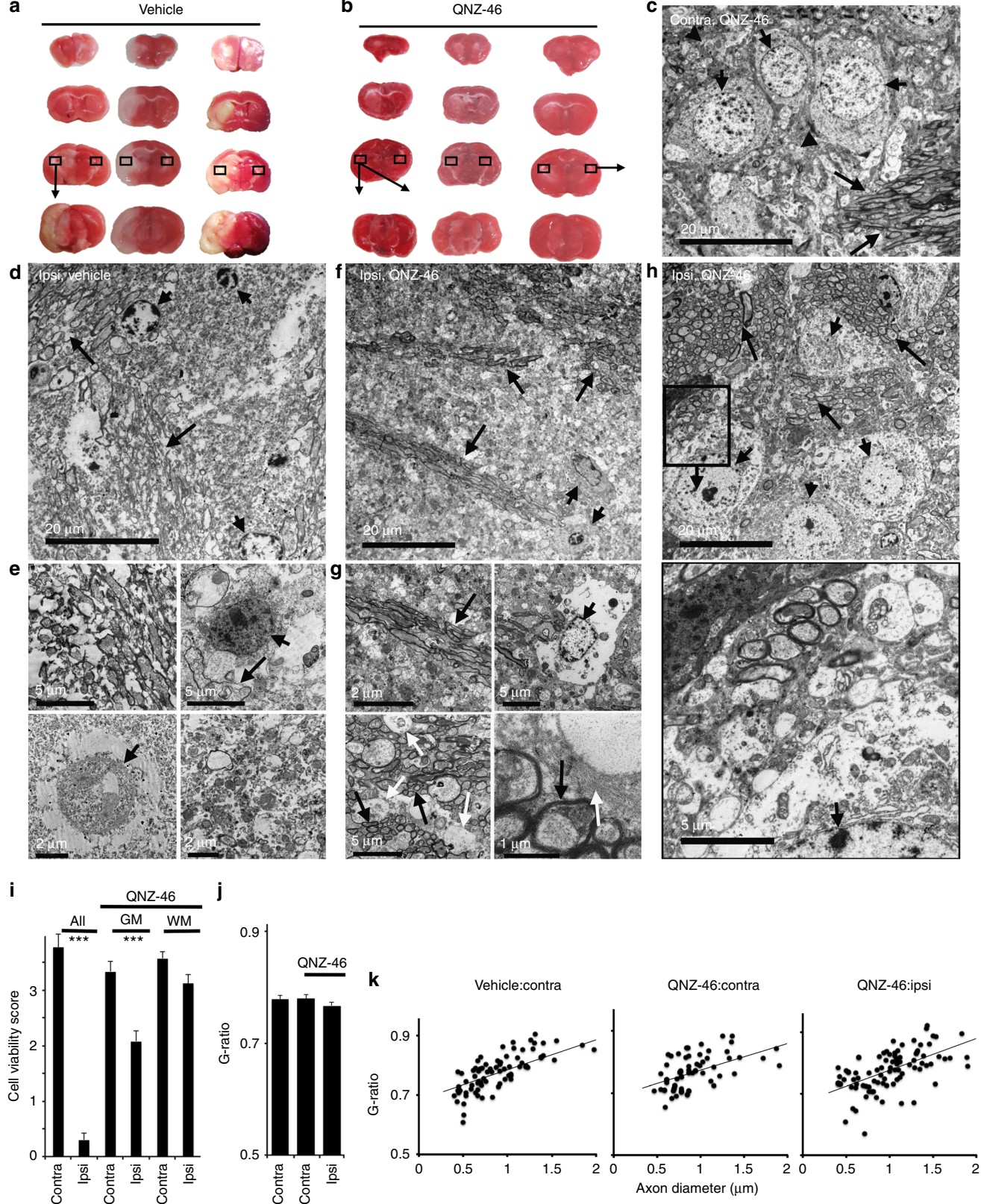

other NMDA GluR subunits[39,40], two additional factors that will limit the side-effects of drug treatment. Stroke incidence is well correlated with age, prior transient ischemic attack, hypertension, and prior stroke[41]. Our findings suggest that prophylactic treatment in patients at risk of stroke may offer an alternative strategy for clinical intervention that avoids the therapeutic window. The extent of the structural and functional neuroprotection offered by QNZ-46 is surprising; in particular the effect in gray matter where neuron somata and neuropil were protected against ischemic injury in vivo. Targeting of extra-synaptic NMDA receptors has shown promise for a wide range of neurological conditions including dementia[42], and QNZ-46 may protect gray matter in a similar fashion. The combined protective effect in both categories of CNS tissue adds to the clinical potential of the drug.

## Methods

**Animals and reagents**. All animal procedures conformed to local ethical standards and ARRIVE guidelines. UK home office and Maltese national regulations were followed as appropriate. For RON experiments, nerves were dissected from juvenile (P8–12) or adult (P90–120) Wistar rats, adult (P80–110) PLP-GFP or C57 wild-type mice. Acute coronal vibratome-cut sections were cut in oxygenated, ice-cold cutting solution (in mM): NaCl, 92; KCl, 2.5; NaH$_2$PO$_4$, 1.2; MgSO$_4$, 2; CaCl$_2$, 2; NaHCO$_3$, 30; glucose, 25; Hepes, 20; Na Pyruvate, 3; Thiourea, 2; Na Ascorbate, 5; pH, 7.4. Experiments were performed in artificial cerebrospinal fluid (aCSF), composition (in mM): NaCl, 126; KCl, 3; NaH$_2$PO$_4$, 2; MgSO$_4$, 2; CaCl$_2$, 2; NaHCO$_3$, 26; glucose, 10; pH, 7.45, bubbled with 5% CO$_2$/95% O$_2$ and kept at 37 °C. KCl (47 mM) was added to aCSF (NaCl replacement) for "High K$^+$" experiments. For zero-Ca$^{2+}$ aCSF, CaCl$_2$ was omitted, and 50 μM EGTA was added. For zero-Na$^+$ aCSF, Na$^+$ was replaced with NMDG. Oxygen–glucose deprivation was used as the model of ischemia for CAP and confocal imaging experiments: aCSF was replaced by a glucose-free aCSF (+10 mM sucrose to maintain osmolality) saturated with 95% N$_2$/5% CO$_2$. The chamber atmosphere was switched to 5% CO$_2$/95% N$_2$ during OGD perfusion. Osmolarity of solutions was measured and adjusted as required. NBQX was purchased from Tocris (UK), PPDA and QNZ-46 from either Tocris or Abcam (UK), bafilomycin from Viva Bioscience (UK); all other reagents were from Sigma (UK) including carbenoxolone (CBX), 5-nitro-2-(3-phenyl-propylamino)benzoic acid (NPPB), sulfasalazine (SAS) and threo-beta-benzyloxyaspartate (TBOA).

**In vivo treatment for in vitro recording**. Adult C57 mice were injected i.p. with 200 μl of 50% DMSO solution containing 1 mM β-cyclodextrin +20 mg/kg QNZ-46, or a vehicle control without the QNZ-46. Injections were performed blind by animal house staff and mice were left for 240 min on a warming pad. Other than mild sedation in both groups attributed to the DMSO, the animals showed no signs of distress or aberrant behavior. RON or brain slices were collected as above, in solutions that did not contain QNZ-46.

**Electrophysiology**. RON CAPs were evoked and recorded with glass electrodes and the rectified area used to determine changes in conduction. CAPs were evoked via square-wave constant current pulses (Iso stim A320, WPI), amplified (Cyber Amp 320, Axon Instruments), subtracted from a parallel differential electrode, filtered (low pass: 800–10,000 Hz), digitized (1401 mini, Cambridge Electronic Design) and displayed on a PC running Signal software (Cambridge Electronic Design). Non-recoverable CAP loss from the RON indicates irreversible failure of axon function. Glutamate microelectrode biosensors (Sarissa Biomedical, Coventry,

UK), amplified via a Duo-Stat ME-200+ potentiostat (Sycopel International, London, UK), were used to record glutamate concentration. Signals were differential to a null electrode and both active and null electrodes were gradually inserted into brain slice or RON, in the latter case through a small incision in the nerve sheath. An Ag/AgCl reference electrode was introduced into the bath. Sensors were calibrated in the chamber at the end of each experiment. Values from the null, sensor, and sensor-minus-null outputs were recorded at 0.5 Hz and subsequently converted into glutamate concentration (Supplementary Fig. 4). Recorded glutamate was high following electrode placement, presumably due to localized tissue damage (Supplementary Fig. 4), and declined to a low stable concentration over a 120–180 min (neonatal RON), 300–420 min (adult RON) or 180–240 min (slices) rest period before experiments were initiated. Nerves may have been pre-treated with 50 μM BAPTA-AM for zero-Ca$^{2+}$ experiments (no difference was detected and data were pooled). Corpus callosum glutamate sensor experiments were conducted on 400 μm sections from adult Wistar rats, adult (P80–110) THY-1/YFP (Line H), GFAP-GFP or C57 wild-type mice. Slices were gradually warmed to 37 °C over ~60 min and rested for 120 min prior to use. Unless otherwise stated, preparations were maintained in an oxygenated (1.5 l/min) interface perfusion chamber (Harvard Apparatus Inc.) and continuously superfused (0.6–1 ml/min) with aCSF. Due to the oxygen-sensitivity of the glutamate biosensors, chemical anoxia (1 nM rotenone for neonates/25 μM antimycin-a for adults) + aglycemia was used as the model of ischemia for [glutamate]$_e$ recordings.

**Two-photon confocal imaging**. After deep isoflurane anesthesia and decapitation, the brain was rapidly removed into chilled aCSF supplemented with 75 mM sucrose and vibratome sectioned into coronal slices (400 μm thick) from the genu of the corpus callosum through the caudal extent of the hippocampus. Immediately after sectioning, slices were transferred to a Haas-type interface brain slice chamber (Harvard Apparatus, South Natick, MA) and allowed to recover at room temperature in aCSF for 60 min. Slices were transferred to a mini submerged chamber (0.5 ml) with a coverglass bottom (Warner Instrument Corporation, Hamden, CT) mounted on an upright BX50W1 Olympus Multiphoton microscope (Olympus, Tokyo, Japan) and perfused with room temperature aCSF at 3.5 ml/min. Final temperature control (37 ± 1 °C) was maintained using an in-line heater (Warner Instrument Corporation, Hamden, CT) equipped with a feedback thermistor placed in the chamber and the temperature raised gradually over 60 min. The multiphoton system housed Keplerian beam expanders with IR introduction light paths. A mode-locked MaiTai HP DeepSee laser system (Spectra-Physics) with a tuneable Ti: sapphire oscillator (690–1040 nm) was used as the excitation light source (pulse width < 100 fs; pulse repetition rate 80 Mhz) and controlled through an acousto-optical-modulator. The Group Velocity Dispersion was electronically compensated by a prism-coupled pre-chirper and the beam diameter adjusted by a Keplar telescope. Image acquisition was performed using the Olympus FluoView software.

Vesicular imaging was conducted on brain slices from transgenic animals using FM4-64. Corpus callosum slices were initially superfused with aCSF + 10 μM FM4-64 for 10 min. Slices were then subjected to a 50 mMK$^+$ aCSF + 10 μM FM4-64 for 5 min, and subsequently returned to aCSF + 10 μM FM4-64 for a further 20 min. Next, slices were washed in aCSF (without FM4-64) for 15 min and a suitable region of the corpus callosum was identified. Finally, slices were exposed to 50 mMK$^+$ aCSF for 10 min (or ischemia for 30 min) to promote vesicular fusion/FM4-64 unloading. Images were acquired every 10 s following laser excitation at 890 nm and collected using standard red and green filter settings. Mean pixel intensity within YFP(+) axons or GFP(+) astrocytes was determined using Olympus FluoView software, with the regions of interest determined by the green profile of axons/glia. Fluorescent emission scans during excitation at 405 nm were conducted using 10 nm bin width and the lambda-scan function, with 10 mM QNZ-46 in DMSO diluted 50:50 in immersion oil.

**Fig. 8** Structural protection following systemic QNZ-46 pre-treatment. **a**, **b** Brain lesions in vehicle- and QNZ-46-treated mice subject to the standard MCAO protocol. Boxed areas (external capsule-motor cortex border) were examined via TEM. **c** Cortical neuronal somata (nuclei indicated by short arrows), neuropil (arrow heads), and myelinated axons (long arrows) in contralateral QNZ-46-treated mice appeared normal with no unusual features. **d**, **e** Ipsilateral vehicle-treated external capsule-motor cortex border showed wide-scale cellular destruction and loss of structure including disrupted myelinated axon tracts (arrows **d**, **e** top left) and necrotic neuronal soma (arrow heads **d**, **e** bottom left). White matter glial cell soma and process were also necrotic (**e**, top right, short arrow) and occasional distorted myelin profiles were apparent (**e**, top right, arrow). Cell processes in the neuropil were generally disrupted with free-floating mitochondria present (**e**, bottom right). **f–h** Ipsilateral QNZ-46-treated external capsule-motor cortex border showed almost normal structural features in both white matter and gray matter areas. Myelinated axon tracts showed no myelin damage (**f**, **g** top left, **h** arrows) and both neuronal and glial soma (**f**, **g** top right, **h** short arrows) appeared largely intact. Glial processes were often swollen (**g**, bottom left and right white arrows) and in many cases could be identified as astrocytic containing glial filaments (**g**, bottom right, white arrow). Neuronal, neuropil, and myelinated axon profiles are shown at higher gain in the expanded boxed area in **h** (lower panel). **i** Cell viability was severely compromised (ANOVA with Holm–Šídák post test; ***$P$ = 0.0000) in ipsilateral vehicle-treated mice (where white matter and gray matter, neuron and glia could not always be distinguished and are grouped together). QNZ-46 pre-treatment significantly improved cell viability in both white matter and gray matter (***$P$ = 0.0000 vs. untreated ipsilateral injury), with no significant difference between ipsi- and contralateral cells in white matter regions. **j–k** G-ratio was not significantly different in ipsi- and contralateral white matter axons in QNZ-46 pre-treated mice. Note axon disruption precluded the measurement of G-ratio in ipsilateral vehicle-treated white matter

**Single-photon imaging**. Live imaging of NMDA-receptor-mediated myelin injury: PLP-GFAP RONs were exposed to FluoroMyelin Red (5%; 100 min, 5 °C in cutting solution) with or without drug, prior to dye wash-out and mounting in a temperature-controlled perfusion chamber (Warner Instruments, Hamden, CT, USA). RONs were superfused at 2 ml/min with aCSF with or without drug and imaged in a single plane on an inverted Nikon TE2000-U microscope. Imaging was achieved via spinning disk confocal imaging (Crest-Optics, X-light) and collected via MetaMorph software (Molecular Devices). FluoroMyelin Red and GFP was consecutively imaged following excitation of both at 488 nm using a $650 \pm 50$ nm BP filter (Thorlabs, Newton, New Jersey, USA) and a standard GFP filter set (Chroma Technology Corporation, Bellows Falls, VT, USA), respectively. RONs were imaged 20 min prior to switching to a 60-min period of OGD and then switched back for 60 min of recovery. Myelin decompaction was assessed using intensity line-plots drawn perpendicular to the longitudinal axis of individual myelin sheaths using ImageJ (NIH). Pre-OGD myelin sheath width was not significantly different in control and treated preparations and the data were pooled. Single-photon, laser scanning confocal images of QNZ-46 bath-loaded adult mouse ON and 200 μm coronal brain sections (P90–120) Wistar rats, (P80–110) PLP-GFP or C57 mice and brain sections from QNZ-46-injected mice, were collected using a Leica TCS SP8 microscope, using loading protocol and settings for GFP and FluoroMyelin Red comparable to those used for spinning disk above. QNZ-46 was imaged following excitation at 405 nm using filter settings standard for DAPI emission. Wide-field QNZ-46 imaging was performed using an Olympus epi-fluorescence microscope and a standard DAPI filter set; emission was quantified from the whole nerve section using ImageJ.

Measurement of myelinic vesicular-mediated Ca2+ increase: brain sections were prepared from adult (p100-p160) CD1 WT mice and dye loaded at room temperature (22 °C) for 2 h in the presence or absence of bafilomycin in continuously bubbled aCSF ($5\% \ CO_2/95\% \ O_2$) containing 10 μM X-Rhod-1 AM and 1 μM $DiOC6^8$. Subsequently, sections were maintained in DiOC6-containing aCSF with or without bafilomycin until imaging. Dye-loaded sections were placed in a perfusion chamber and continuously superfused at 1 ml/min with bubbled aCSF or high-K aCSF (50 mMK+, see above). Single-photon laser scanning confocal images of corpus callosum myelinated axons were collected using a Leica TCS SP8 microscope at 37 °C. Excitation/emission wavelengths was 488/492–560 nm for DiOC6 and 561/585–655 nm for X-Rhod-1. Polygonal ROIs of DiOC6 and X-Rhod-1 dual-loaded axons were selected to exclude adjacent X-Rhod-1-loaded cell bodies. X-Rhod-1 intensity was normalized to the pre-high-K signal and data from individual mice were averaged for a single $n$ (1–3 slices per mouse).

**Transient middle cerebral artery occlusion**. Eleven male Thy1-YFP and 15 C57 wild-type mice weighing 25−30 g were analgesized with buprenorphine (0.03 mg/kg b.w. i.p.) 2 h before surgery and subsequently anesthetized with isoflurane (3% initial, 1.0–1.5% maintenance) and 60% NO in $O_2$. Animals were maintained normothermic (37 ± 0.5 °C) by means of a servo controlled heating blanket (Harvard Apparatus, Holliston, MA) with rectal temperature monitoring. Pulse oximetry (Spo2), heart rate, and respiratory rate were monitored continuously (STARR Life Sciences Corp., Allison Park, PA), along with systemic blood pressure via a Kent CODA® Standard tail-cuff blood pressure system (Kent Scientific Corporation, Torrington, USA). A fiber-optic probe (VP10M200ST, Moor Instruments Ltd, Axminster, UK) was affixed to the skull over the middle cerebral artery for measurement of regional cerebral blood flow using a moorVMS-LDF Laser Doppler System. Under microscope, the left common carotid, internal carotid, and external carotid arteries were exposed through a midline neck incision. The proximal portions of the left common carotid and the external carotid arteries were ligated and a 6–0 silicon-coated nylon suture (60SPRePK5-21910, Doccol Corporation, Massachusetts, USA) introduced into the internal carotid artery to occlude the middle cerebral artery at its origin. Mice were allowed to recover from anesthesia in a warm recovery cage throughout the duration of occlusion. After 60 min, reperfusion was obtained by withdrawal of the suture. Middle cerebral artery occlusion was confirmed by a sudden drop in relative cerebral blood flow to approximately 85% less than baseline measurement. Animals were subsequently administered buprenorphine for analgesia and placed in a recovery cage for 2 h before returning to their home cages with free access to food and water. Plasma glucose concentration was measured before tMCAO, 5 min and 24 h after reperfusion. Sham-treated animals received all surgical procedures but the filament was not inserted into the MCA. For animals pretreated with either QNZ-46 or vehicle (50:50;DMSO/β-cyclodextrin), the active drug or carrier was administered intraperitoneally 120 min before tMCAO.

**Clinical evaluation**. Body weight (g) was recorded prior to surgery and at 24 h for sham-, vehicle- and drug-treated groups. While a slight weight increase was observed in sham controls, experimental groups demonstrated weight loss after 24 h. Ischemia-induced weight loss was attenuated in drug-treated mice compared to vehicle group, although this was not significant (vehicle vs. drug-treated group: 5.30 ± 0.38 g vs. 4.58 ± 0.33 g).

**Modified Bederson score**. Neurological performance[43] was assessed by two independent and blinded investigators 24 h post tMCAO according to the

following scoring system: 0, no neurological deficit; 1, forelimb flexion; 2, decreased resistance to lateral push; 3, unidirectional circling; 4, longitudinal spinning; 5, no movement.

**Corner test**. The corner test detects abnormalities of sensory and motor function including vibrissae, forelimb and hindlimb use, and postural motor function[44]. The apparatus consisted of two cardboard boards each with a dimension of $30 \times 20 \times 1$ cm³. The edges of the two boards were attached at a 30° angle with a small opening along the joint to encourage entry into the corner. The mouse was made to enter between the two boards facing the corner. As the animal progressed into the corner, both sides of the vibrissae were stimulated together and the mouse reared forward and upward, and then back to face the open end of the boards. The direction towards which the mouse turned was recorded for a total of ten trials per animal. Recordings were performed at baseline (prior to surgery) and again at 24 h reperfusion.

**Wire hanging test**. Motor function was evaluated using a wire hanging test for grip strength, balance, and endurance. This test is based on the latency of a mouse to fall off a metal wire upon exhaustion. The apparatus consisted of a 2-mm-thick metallic wire stretched between two poles held 50 cm above the ground, with a pillow in between to prevent injury upon falling. The mouse's hindlimbs were covered with adhesive tape to prevent the animal from using all four paws. The mouse was trained to suspend its body by holding on to the wire. Pre-training was performed for 2 consecutive days before and on surgery day, prior to obtaining baseline values. The mouse was suspended for short intervals (10–20 s) for several trials and then returned to its cage. If the mouse fell before the end of the trial, it was immediately returned and allowed to grasp the wire. Mice were only returned to their home cage once training was completed. Experimental recordings were taken at baseline (prior to surgery) and again at 24 h reperfusion. Suspension time was measured for three trials per session with a maximum of 60 s per trial and a 30-s recovery period in between.

**Infarct volume**. Lesion volume was assessed 24 h post-MCAO following neurological assessment. Brains were removed, sectioned at 2 mm using a brain matrix and incubated in 2% 2,3,5-triphenyltetrazolium chloride (TTC; Sigma-Aldrich) in saline for 20 min at 37 °C. Four slices per mouse were imaged and processed for infarct volume evaluation by blinded manual area tracing using digital imaging and image analysis software (ImageJ). For each section, the unstained area was defined as the ischemic lesion while red-stained areas delineated viable tissue[43]. Direct infarct volume was calculated by linear trapezoidal extrapolation (Cavalieri principle); for each brain sample, volume of infarction was calculated by integrating the area of damage at each stereotactic level and the slice thickness. An indirect/corrected infarct volume was then calculated to compensate for the space-occupying effect of brain edema. For each brain, an edema index was calculated by dividing the sum of ipsilateral hemisphere volumes by the sum of contralateral hemisphere volumes. The actual infarct volume adjusted for edema was then determined by dividing the direct infarct volume by the edema index[45].

**Tissue processing and histological assessment**. Thy-1-YFP mice subject to sham-, vehicle- and drug-treated tMCAO were killed (ketamine/xylazine), transcardially perfused (4% paraformaldehyde in PBS) and the brains were removed, post-fixed (24 h at 4 ºC), cryopreserved (30% sucrose for 72 h at 4ºC), embedded (low-melting point agarose, Sigma-Aldrich, A0576), and chilled. Free-floating 20 μm coronal sections (compresstome VF-300, Precisionary Instruments Inc., San Jose, California) were mounted onto gelatin-coated slides and YFP-imaged (EVOS Auto FL, ThermoFischer Scientific) using the appropriate green (496/518) filter set. The sections were immediately subsequently stained with 0.1% luxolfast blue (LFB, Solvent Blue 38, Sigma-Aldrich)/0.25% cresyl violet (Sigma-Aldrich) and myelin-imaged bright field. The mean optical density (OD) of the LFB stain was used as a measure of myelin integrity[46]. For each section, two ×20 and five ×60 non-overlapping images from homologous ipsilateral and contralateral external capsule were acquired and processed for OD using ImageJ software. OD calibration was carried out such that pixel values, usually in gray level units, were in OD. Each image was converted to 8-bit and the ROI was outlined manually. Any incidental border region of the cortex was excluded from analysis. OD was measured from ipsilateral and corresponding contralateral hemispheres and expressed as a ratio.

**Electron microscopy**. For brain sections: 24 h following the standard tMCAO protocol mice were anesthetized and transcardially perfused for 5 min with cold 4% PFA in 0.1 M Sørensen's solution. The brain was then removed, sectioned at 2 mm using a brain matrix and incubated in 2% TTC for 20 min and assessed for lesion volume as above. Sections were then immersion-fixed in 2.5% gluteraldehyde/0.1 M Sørensen's overnight and stored in Sørensen's solution. The external capsule-motor cortex border region from both hemispheres was excised prior to post-fixation (1% osmium tetroxide), serially dehydration and epoxy infiltration. Ultrathin coronal sections (50–70 nm) were counterstained with uranyl acetate and lead citrate prior to blind examination using a Jeol 100CX electron microscope. Micrographs were analyzed for axon G-ratio by hand tracing (ImageJ, NIH) the external and internal myelin profile of all X-S axons within a micrograph and converting the area to

idealized circles. Somata viability was assessed using a basic scoring system where one point was awarded for each of: (a) an intact cell membrane; (b) the presence of undamaged organelles such as mitochondria; (c) a normal nuclear morphology; and (d) the presence of clear cytoplasm. The external capsule-motor cortex border region of three vehicle-treated and three QNZ-46-treated mice were analyzed.

For RON: following either 30 min OGD (for long-sectioning: L-S) or 60 min OGD + 60 min recovery (for cross-sectioning: X-S), adult mouse nerves were immersion-fixed in 4% PFA in Sørensen's for 5 min followed by 2.5% gluteraldehyde/0.1 M Sørensen's overnight and storage in Sørensen's. A minimum of four grid-sections in each of a minimum of three nerves were analyzed blind by hand including the tracing of axons and myelin profiles (X-S), axoplasmic vesicle counting (L-S) and focal myelin damage (L-S). G-ratio in X-S was calculated as above; vesicle counting was performed in L-S to allow vesicles to be distinguished from microtubules. Focal myelin injury was scored: 0 = normal compact myelin; 1 = one layer of myelin splitting; 2 = myelin bubbling involving multiple lamella; 3 = complete myelin breakdown.

**Statistics**. Data are mean ± SEM, significance determined by $t$ test or ANOVA with Holm−Šídák post hoc test as appropriate; Mann−Whitney $U$ test was performed for nonparametric variables (Modified Bederson Score). CAP and biosensor recordings were stable over long periods of control recording and glutamate receptor antagonists had no effect under control conditions (Supplementary Fig. 6). For long periods of drug pre-treatment, control experiments were performed with the identical protocols to allow direct comparison. *$P$ = <0.05 (rounded three decimal places), **$P$ < 0.01 (rounded three decimal places), ***$P$ < 0.001 (rounded four decimal places). Sample sizes for experimental groups are based on power calculations using established variability. Data from all completed experiments are included and no outliers were excluded. Two animals subjected to tMCAO died post-operatively and were not counted. Sham and test tMCAO trials were alternated and the experimenters were blinded to the contents of the injection; the micrographs generated from these experiments were analyzed blind (REF) and only unblinded once analysis was complete. All test experiments were intercalated with the relevant controls and where possible trials were conducted blind, including all in vivo treatments.

**Data availability**. All relevant data are available from the authors upon request.

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

## Acknowledgements

We thank Robert Zammit for assistance with the multi-photon confocal imaging and Waldemar Woznica for assistance with IP injections performed at Plymouth. This work was supported by BBSRC (J016969/1), by the University of Plymouth and by the Alfred Mizzi Foundation, Malta.

## Author contributions

S.D. performed and analyzed all CAP, biosensor and two-photon microscopy recordings; D.B.H. performed and analyzed the laser-scanning and spinning disk confocal imaging; J.V. conducted the tMCAO protocol; P.B. and G.H. prepared the EM sections; M.V. supervised and contributed to the two-photon imaging and the tMCAO experiments; C. Z. contributed to two-photon imaging; R.F. supervised the project, developed the theory of a myelin shield, analyzed the EM, and wrote the manuscript.

## Additional information

**Competing interests:** The authors declare no competing interests.

