## [Peer review file · Nature Communications]

Reviewers' comments:

Reviewer #1 (Remarks to the Author):

The mechanisms mediating myelin damage in diseases such as MS and stroke are currently ill defined. Previous studies have demonstrated the presence of NMDA receptors at high levels in myelin and activation of those receptors is thought to be detrimental to myelin integrity. Past work has suggested that glutamate released from glial cells in the surrounding environment mediates such damage. In the current manuscript, the authors propose an alternative model in which vesicular release of glutamate from axons triggers a perturbation of myelin integrity and that this disruption can be blocked by a negative allosteric modulator of the GluN2C/D NMDA receptor. There are some interesting aspects to the manuscript, which may provide novel insights into myelin damage in a variety of settings and neural conditions. However, there are significant concerns with the data as presented. Two major issues that are of concern:

1) First, it is not clear from the imaging data in Figure 1 that there is a high level of vesicular glutamate in axons. The data that glutamate release is driven by vesicle fusion and not other methods is indirect but relatively strong. However, the localization to axons, in particular vesicles, is somewhat weak. This would be strengthened if the authors were to demonstrate vesicular fusion in response to a depolarizing activation in a more precise manner.

2) The 2nd major concern relates to the observation that damage results in perturbation of myelin integrity. First, it is unclear that vesicular released from axons will reside in the periaxonal space under the myelin sheath as stated, but rather may be targeted to Nodes of Ranvier or other unmyelinated areas. Second, the data demonstrating a perturbation of myelin is equivocal. Data in figure d is somewhat hard to interpret but suggests that oligodendrocyte processes but not myelin are disrupted and the data in Figure 3e is completely incomprehensible. Indeed, the only data showing compelling myelin sheaths is the incorporation of QNZ-46 shown in Figure 4. It may be more convincing to show cross sectional images of treated optic nerves.

It would seem to be essential to support the authors argument that some ultrastructural studies are conducted to demonstrate timing and dose relationship of myelin perturbation. Indeed, currently it is not evident that the direct target of glutamate release is the myelin sheaths rather than axonal dysfunction, perhaps mediated by hyper-excitability. Along the same lines the data fails to exclude the possibility that axonal damage may precede myelin breakdown. While there are some interesting aspects to this study, including the potential to mitigate damage through modulating glutamate receptors activity with QNZ-46, the mechanism and target of interaction remains somewhat uncertain and requires additional studies.

Reviewer #2 (Remarks to the Author):

This manuscript expands prior observations (Micu 2006) that energy deprivation damages CNS white matter by toxic accumulation of glutamate and activation of myelinic NMDA receptors. Using pharmacological approaches with in vitro models of acute white matter brain slices and rat optic nerves, the primary new observations are that glutamate release may occur through vesicular release (rather than reverse glutamate transport) and that NMDA receptors containing NR2C/D subunits may be differentially important for myelin injury. The study takes advantage of an antagonist, QNZ-46, recently reported to be a lipophilic allosteric modulator selective for these NR2 subunits. In vivo, QNZ-46 pretreatment reduces grey and white matter infarct size and improved behavioral deficits following transient MCAO ischemia. Additionally, there is an important experimental observation that some NMDA antagonists (including QNZ-46 and MK-801) are slow to penetrate myelin, and may require extensive pretreatment before exerting pharmacological blockade.

This is a comprehensive study which uses a range of optical and physiological approaches. It

meaningfully advances understanding of white matter injury mechanisms.

Major comments:

- 1) Given the new observation that commonly used glutamate blockers require extensive wash-in, some caution may be appropriate in interpretation of negative results using other drugs. Are the authors confident that the same is not true for TBOA?
- 2) What is the evidence that FM4-64 (vesicular uptake) was confined to axons, and not on the inner myelin membrane? Can these structures be distinguished at the level of light microscopy?
- 3) The manuscript provides proof-of-concept support for the idea that selectively targeting NMDA receptor subunits found in both gray matter and white matter myelin may be effective for stroke neuroprotection. The lipophilic compound QNZ-46 is probably poorly suited for acute intervention because of its slow penetration, although this could be tested directly using post-stroke treatment models. It is probably a stretch to suggest that these findings support rational design of long-term prophylactic therapy for patients at risk for stroke. (The main concern would be behavioral or cognitive adverse effects, which are not studied.)

Minor comment:

- 1) Is it correct that extracellular glutamate is recorded in millimolar rather than micromolar concentrations? Similarly some experimental drug concentrations (eg TBOA) are probably misstated as millimolar.

Reviewer #3 (Remarks to the Author):

The mechanisms of ischemia-induced white matter damage and strategies to protect it, are important issues with translational relevance. The pathway that the authors describe is novel and of interest, but there are currently too many assumptions and indirect observations underpinning their model as it stands.

-the authors need to prove directly what they claim in the abstract i.e. that myelinic 2C/D NMDARs are activated following release of glutamate into the periaxonal space:

-the authors should show direct axonal release of glutamate into the peri-axonal space. If myelinating oligodendrocytes were made to express GluSNFR then they could detect this

-the authors should show the extent to which oligodendrocytes have NMDAR currents and their sensitivity to QNZ-46, and other antagonists such as DQP-1105. They need to show that they are activated following ischemia by patch clamping or calcium imaging.

To be translationally relevant as claimed in the abstract, appreciable currents or expression needs to be confirmed in human oligodendrocytes.

-given high expression of GluN2C in the cerebellum, an expanded battery of tests involving this circuitry should be provided to confirm tolerance of this drug.

It is unclear to me how a 2C/2D antagonist putatively protecting myelin should have the degree of effect on total lesion size shown in Fig. 5b. Is DQP-1105 equally effective? Confirmation of protection by genetic means such as knock-down of Grin2c should be attempted.

-the figure legends are far too brief and very unhelpful-what little text there is simply reiterates the conclusions in the results section. There is no specific statistical information in the legends nor quoting of p values, and little explanation of the experiment.

-In 1i saying the Glu release is 'insensitive' to TBOA is not really correct-more accurately it is not inhibited-TBOA is increasing glutamate levels presumably by preventing re-uptake.

-Is 1j mislabelled (should be bafilomycin not TBOA?)

Reviewer #4 (Remarks to the Author):

In this study, Doyle et al., re-envisage the role of glutamate and glutamate receptors in white matter ischemia, and determine the release mechanism of glutamate in the white matter. The authors elegantly show, using a combination of electrophysiology and imaging, that in ex-vivo glutamate concentrations increase in the white matter during high potassium application (mimicking high neuronal activity) and in ischemia, and blocking NMDA receptors increases action potential propagation recovery from ischemic insults. The authors clearly demonstrate that glutamate is released in the white matter and argue that this is via a vesicular release in myelinated axons into the periaxonal space both during high potassium application and in ischemia; thus, confirming Micu et al., 2016 finding (Micu et al., *Experimental Neurology* 2016; 276, 41–50). The authors further identify a novel NMDA- GluN2C/D subunit blocker that significantly reduces ischemic white matter damage in ex vivo and in vivo, which might be therapeutically significant, thereby reawakening the notion of therapeutic NMDA receptor blockage for therapy after ischemia. Therefore, this study is of considerable interest, particularly due to the therapeutic potential for white matter ischemia. However, regrettably, the study does not sufficiently address the recent papers questioning the role of NMDA receptors and/or glutamate release in white matter ischemia. To further address these papers and to strengthen the conclusions made within the manuscript further evidence is needed, as listed below.

Major comments

(1) The role of glutamate and specifically NMDA receptors in the white matter has been under considerable debate over the last 10 years, and in recent years two major papers cast doubt on previous reports. Saab et al, showed that knocking out NMDA receptors in oligodendrocytes made white matter ischemia outcome worse, not better (Saab et al., 2016 *Neuron*, 91(1), 119–132.) and Hamilton et al., argued that the main damage in white matter ischemia is mediated via TRPA1 channels, not NMDA receptors and glutamate release (Hamilton et al., 2016 *Nature*, 529, 523–527). The authors need to better address or discuss, the differences between this report, that is in line with earlier reports, and recently published papers.

Particularly, with regards to TRPA1 channels. In this manuscript blocking TRPA1 channel did not show a functional effect on the recovery of the propagation of the action potential in the RON, nor did it prevent ischemia-induced swelling of myelin, whereas in Hamilton et al., (*Nature* 2016), blocking TRPA1 channels significantly prevented myelin lamella separation seen after ischemia (see also comment (2)).

The authors did note that a short pre-incubation with NMDA receptor blocker MK801 might explain the difference between previous reports and this manuscript, although this may account for the difference from Baltan et al. (*J. Neurosci* 2008, 28:1479-1489) and this manuscript, it does not explain the difference of the findings in SAAB et al., (*Neuron* 2016) which did not use blockers but genetically deleted GluRN1 from oligodendrocytes.

These differences need to better discussed and addressed.

(2) In Figure 3f myelin width is measured, and it is not clear to me as to what is being measured, nor how it is measured (not explained in methods). Given the expertise of the group with electron microscopy (EM) of myelin in the optic nerve, I find it surprising that myelin integrity with EM was not measured, which is a more reliable method and would directly address whether there are differences in myelin lamella separation after ischemia as seen in Hamilton 2016 or not, and whether QNZ-46 improves myelin integrity. Similarly, in figure 5g, low mag staining with Luxol fast blue and level of dye uptake was measured by optical density as a readout of myelin integrity, which does not provide conclusive evidence of myelin integrity, this should be done by electron microscopy and myelin integrity quantified.

(3) Figure 1 a-f, and Suppl.Fig. 1d, show an inconsistency of FM4-64 loading, in Figure 1c and Suppl.Fig. 1d top panel, there is punctate staining, whereas in Figure 1d and Suppl.Fig. 1d lower panel it shows a continuous staining. This continuous staining is interpreted as evidence for that the axonal vesicular release occurs under the myelin sheaths into periaxonal space – pg2, line 8 “the majority of vesicular fusion must occur under the myelin sheath and empty into the periaxonal space” and on page 3, line 8 “The absence of focal sites of vesicular fusion in axons indicate that ischemic glutamate release empties into the periaxonal space under the myelin sheath.”

Yet

most experiments using FM4-64 labelling with 2photon microscopes detect punctate staining, and so did Micu et al., (Experimental Neurology 2016) in the myelinated optic nerve. However, this concern of the specificity of the labelling is mitigated somewhat by the axon-specific reduction in staining and the constant level of FM4-64 staining in astrocytes (although how this was quantified is not clearly explained in the manuscript), and the concomitant measured increase in glutamate within in the optic nerve occurring at a similar timeframe. However, concluding that this continuous type of FM4-64 loading is evidence for release under the myelin and into the periaxonal space is somewhat an over interpretation. But given that over 90% of the axon surface is covered with myelin, particularly in the optic nerve, it is thus likely that the release is under the myelin like Micu et al. argued. Nonetheless, clear evidence for vesicular release into the periaxonal space is not provided thus these statements in the manuscript should be left out, or toned down and only speculated on. Whether it is or not is not essential to the core finding of the manuscript thus the authors should rather focus their efforts to clearly demonstrate that the de-staining of FM4-64 is affected by blockers of vesicular release – this would truly strengthen their core finding that vesicular release from axons triggers myelin damage.

(4) The finding that QNZ-46 reduces stroke volume and improves behavior in mice after 60 min of middle cerebral artery occlusion is the major finding of this paper, and the most exciting. Therefore, it is surprising the lack of detail of the behavior in figure 5, and ultrastructural analysis of myelin integrity (as mentioned above). The manuscript would significantly improve if more detail would be given to the behavioral tests and their outcome, and the effect of QNZ-46 on neurons (or grey matter) to strengthen the comments made in the manuscript regarding using QNZ-46 as a prophylactic therapy approach in patients at risk for stroke.

Minor comments

(1) Results section- page2, line 26 “Resting extracellular glutamate was 1.6 +/-1.7 mM in adult CC, and TBOA evoked a 1.3 +/-0.4 mM peak increase..” is confusing. When comparing this data to the data presented in supplementary figure 2, it is not clear as to whether the value 1.6 +/-1.7 mM of resting extracellular glutamate relates to Suppl.Fig. 2a, which in fact shows around 1.6mM of resting glutamate levels, or whether it should relate to suppl. Fig. 2b, where control max change in [glutamate] in ‘resting’ condition before TBOA are given as ~0.5mM, as the max change in glutamate levels in TBOA in that figure is ~1.3mM – same as in the text. Are the authors comparing ‘resting’ [glutamate] levels in Suppl.Fig. 2a to TBOA max change in [glutamate] levels in Suppl.Fig. 2b. – This is confusing and needs better clarification in the text as supplementary

figures 2a and 2b are not showing the same (2a is showing measured levels in [glutamate], and 2b is showing the max change in [glutamate]).

(2) In Figure 2a, the trace is missing for recordings in solutions lacking Na⁺, and in 2b the trace recorded with Diltiazem is missing.

(3) Figure 2- it would be clearer if the bar graph for Rose Bengal and bafilomycin/tboa was labelled as (d), as it presumably it is figure 2d!

(4) Page 3 line 32 - "Myelin sheath diameter increased from 0.67 mm +/-0.02 to 0.90 mm +/- 0.07 after 60 min ischemia + 60 min recovery, an effect that was prevented by 120 min pre-treatment with QNZ-46 but not by TRP1A block (Fig. 3 e, f)" – the unit should be in μm not mm.

(5) There are a number of typos and errors within the text and figures, such as supplementary figure 2a, where RON should be MON, when showing the glutamate concentration for mouse optic nerve, instead of rat optic nerve. - and on page 3, line 3 NPPD should be NPPB, etc..

Reviewer #1:

“The mechanisms mediating myelin damage in diseases such as MS and stroke are currently ill defined. Previous studies have demonstrated the presence of NMDA receptors at high levels in myelin and activation of those receptors is thought to be detrimental to myelin integrity. Past work has suggested that glutamate released from glial cells in the surrounding environment mediates such damage. In the current manuscript, the authors propose an alternative model in which vesicular release of glutamate from axons triggers a perturbation of myelin integrity and that this disruption can be blocked by a negative allosteric modulator of the GluN2C/D NMDA receptor. There are some interesting aspects to the manuscript, which may provide novel insights into myelin damage in a variety of settings and neural conditions. However, there are significant concerns with the data as presented. Two major issues that are of concern:

1) First, it is not clear from the imaging data in Figure 1 that there is a high level of vesicular glutamate in axons. The data that glutamate release is driven by vesicle fusion and not other methods is indirect but relatively strong. However, the localization to axons, in particular vesicles, is somewhat weak. This would be strengthened if the authors were to demonstrate vesicular fusion in response to a depolarizing activation in a more precise manner. 2) The 2nd major concern relates to the observation that damage results in perturbation of myelin integrity. First, it is unclear that vesicular release from axons will reside in the periaxonal space under the myelin sheath as stated, but rather may be targeted to Nodes of Ranvier or other unmyelinated areas.”

In response to the reviewer’s question we have used the Rhod-x-1 myelin imaging technique developed by Stys et al., (Nature, 2006; Nature Medicine 2007) to record the Ca²⁺ rise evoked in peri-axonal myelin when vesicular fusion is stimulated with high-K⁺. The Ca²⁺ rise is shown to be sensitive to block of vesicular release (data in the new Fig. 2). In addition, we now report ultrastructural evidence for axoplasmic vesicle fusion at the sub-myelinic axolemma (new Fig. 2), the structural correlate of release into the periaxonal space. We have also improved the text relating to FM4-64 imaging in myelinated axons. Using this technique, vesicular release is uniform along myelinated axons and since ~99% of these axons are enclosed by the periaxonal space, release must *a priori* occur primarily into the periaxonal space.

“Second, the data demonstrating a perturbation of myelin is equivocal. Data in figure d is somewhat hard to interpret but suggests that oligodendrocyte processes but not myelin are disrupted and the data in Figure 3e is completely incomprehensible. Indeed, the only data showing compelling myelin sheaths is the incorporation of QNZ-46 shown in Figure 4. It may be more convincing to show cross sectional images of treated optic nerves.

It would seem to be essential to support the authors argument that some ultrastructural studies are conducted to demonstrate timing and dose relationship of myelin perturbation. Indeed, currently it is not evident that the direct target of glutamate release is the myelin sheaths rather than axonal dysfunction, perhaps mediated by hyper-excitability. Along the same lines the data fails to exclude the

possibility that axonal damage may precede myelin breakdown.” **We have added an extensive set of EM data, including quantitative analysis of myelin integrity after 30 and 60 min of OGD in the isolated optic nerve preparation (new Fig. 5) and in the contra- and ipsi-lateral external capsule-motor cortex region of the *in vivo* stroke model (new Fig. 8). In addition, we have included an analysis of early myelin pathology after 30 min of OGD, which precedes pathology of the axon cylinder as the reviewer predicted (e.g., new Fig. 2 d). We have also remade the original Fig. 3e (new Fig. 4e), focusing on a smaller area to improve clarity. Regarding the concern about Fig 3d, this panel demonstrates that oligodendrocyte processes are distinct from myelin; there is no injury or disruption in this panel. Regarding the concern that hyper-excitability may affect recovery, we have shown in supplementary Fig. 6 f that the NMDA blocker MK-801 produces no significant change in the CAP.**

“While there are some interesting aspects to this study, including the potential to mitigate damage through modulating glutamate receptors activity with QNZ-46, the mechanism and target of interaction remains somewhat uncertain and requires additional studies.” **An extensive ultrastructure analysis of the *in vivo* stroke model in the presence and absence of QNZ-46 is now included in the manuscript. By focusing on the external capsule-motor cortex border for this analysis we are able to describe the nature of the protection the drug provides to neuronal somata, neuropil, glial cell somata, glial processes, myelin and the axons themselves (new Fig. 8).**

Reviewer #2:

“This manuscript expands prior observations (Micu 2006) that energy deprivation damages CNS white matter by toxic accumulation of glutamate and activation of myelinic NMDA receptors. Using pharmacological approaches with *in vitro* models of acute white matter brain slices and rat optic nerves, the primary new observations are that glutamate release may occur through vesicular release (rather than reverse glutamate transport) and that NMDA receptors containing NR2C/D subunits may be differentially important for myelin injury. The study takes advantage of an antagonist, QNZ-46, recently reported to be a lipophilic allosteric modulator selective for these NR2 subunits. *In vivo*, QNZ-46 pretreatment reduces grey and white matter infarct size and improved behavioral deficits following transient MCAO ischemia. Additionally, there is an important experimental observation that some NMDA antagonists (including QNZ-46 and MK-801) are slow to penetrate myelin, and may require extensive pretreatment before exerting pharmacological blockade.

This is a comprehensive study which uses a range of optical and physiological approaches. It meaningfully advances understanding of white matter injury mechanisms.

Major comments:

1) Given the new observation that commonly used glutamate blockers require extensive wash-in, some caution may be appropriate in interpretation of negative results using other drugs. Are the authors confident that the same is not true for TBOA?” **The reviewer raises a good point, however the glutamate transporters that TBOA act upon are expressed in mature myelinated axons primarily at the node of Ranvier rather than within the internodal axolemma (Arranz et al.,**

2008) and the highly lipid soluble TBOA is therefore likely to have rapid access to axonal glutamate transport. We have updated the text to reflect these facts (page 2, line 28). In further support of this, supplementary Fig. 2 shows the rapid effect of TBOA; a significant rise in glutamate within the initial 5-10 minutes of exposure.

“2) What is the evidence that FM4-64 (vesicular uptake) was confined to axons, and not on the inner myelin membrane? Can these structures be distinguished at the level of light microscopy?” **The profile of FM4-64 fluorescence fell within the envelope of the YFP fluorescence of myelinated axons (Fig. 1 d). We have now also included an ultrastructural analysis of L-S myelinated axons which localized vesicles to the axolemma, none where found in the myelin membrane or glial processes (Fig. 2 c-e).**

3) “The manuscript provides proof-of-concept support for the idea that selectively targeting NMDA receptor subunits found in both gray matter and white matter myelin may be effective for stroke neuroprotection. The lipophilic compound QNZ-46 is probably poorly suited for acute intervention because of its slow penetration, although this could be tested directly using post-stroke treatment models. It is probably a stretch to suggest that these findings support rational design of long-term prophylactic therapy for patients at risk for stroke. (The main concern would be behavioral or cognitive adverse effects, which are not studied.)” **The reviewer is correct that drugs acting at GluN2C/D subunits may have behavioral effects but we have argued that these effects are likely to be less than drugs acting at GluN2A/B subunits which are more commonly found at synapses, while the negative allosteric mode of action and use-dependent features of QNZ-46 should target pathogenic receptor activation over physiological receptor activation. We have made these points clearer in the text (last two paragraphs of the discussion).**

“Minor comment:

1) Is it correct that extracellular glutamate is recorded in millimolar rather than micromolar concentrations? Similarly some experimental drug concentrations (eg TBOA) are probably misstated as millimolar.” **We apologize for these typos which has been corrected.**

Reviewer #3:

“The mechanisms of ischemia-induced white matter damage and strategies to protect it, are important issues with translational relevance. The pathway that the authors describe is novel and of interest, but there are currently too many assumptions and indirect observations underpinning their model as it stands.

-the authors need to prove directly what they claim in the abstract i.e. that myelinic 2C/D NMDARs are activated following release of glutamate into the periaxonal space:” **We have now included in the manuscript an analysis of the Ca²⁺ elevation in peri-axonal myelin that follows depolarization: an event that is shown to be blocked by bafilomycin and is therefore a product of vesicular**

release. We have employed the rhod-x-1 myelin imaging technique developed by Stys et al., (Nature, 2006; Nature Medicine 2007) which selectively records Ca^{2+} in peri-axonal myelin. In addition, by fixing tissue at the point of maximal glutamate release during OGD we show vesicular fusion to the sub-myelinic axolemma at the ultrastructural level. We believe that these findings address the reviewers concerns and significantly strengthen the manuscript.

“-the authors should show direct axonal release of glutamate into the peri-axonal space. If myelinating oligodendrocytes were made to expression GluSNFR then they could detect this” **The reviewer makes an excellent and expert suggestion. However, our experience suggest that fluorescent protein expression in oligodendrocytes penetrates only very poorly into the inner myelin layers (for example see new Fig. 4 d). Furthermore, the inner myelin layer is only 140 nm thick and fluorescent changes in this structure could not be resolved from those in outer myelin layers. For these reasons we adopted the well-characterized technique developed by the Stys laboratory which employs dual loading of DiOC₆(3) and X-Rhod-1 selectively into the inner myelin layer to demonstrate myelinic calcium rises in response to vesicular release. We have also captured the structural correlate of this event, targeting of vesicles to the sub-myelinic axolamma.**

“-the authors should show the extent to which oligodendrocytes have NMDAR currents and their sensitivity to QNZ-46, and other antagonists such as DQP-1105. They need to show that they are activated following ischemia by patch clamping or calcium imaging.” **These are excellent suggestions, however it is very likely that currents in the inner myelin layer will be electrically isolated from recordings made from the oligodendrocyte soma, that is after all the function of the myelin sheath.**

“To be translationally relevant as claimed in the abstract, appreciable currents or expression needs to be confirmed in human oligodendrocytes.” **NMDA receptor expression has recently been confirmed in human myelin and oligodendrocytes (Christensen et al 2016), as now stated in the text (page 1, line 30).**

“-given high expression of GluN2C in the cerebellum, an expanded battery of tests involving this circuitry should be provided to confirm tolerance of this drug.” **Toxicological and behavioral testing of this class of drugs is currently being conducted but will not form part of this manuscript.**

“It is unclear to me how a 2C/2D antagonist putatively protecting myelin should have the degree of effect on total lesion size shown in Fig. 5b. Is DQP-1105 equally effective? Confirmation of protection by genetic means such as knock-down of Grin2c should be attempted.” **Having conducted a second series of tMCAO experiment, we now include an extensive ultrastructural analysis of the protected region. We have not attempted the GluN2C KD experiment suggested by the reviewer since protein exchange rates in myelin are very slow (often with half times measured in months) and KD is likely to have an equally slow/partial effect upon myelin injury sensitivity. There is also a concern that GluN2D expression may compensate meaning that interpretable data is**

unlikely. Similarly, Saab et al (2016) recently demonstrated that NMDA receptors regulate glucose import and axonal energy metabolism. Thus, KD of 2C/D may lead to a disruption in energy metabolism and pathogenic mechanisms in white matter independently of the acute events described in our study.

“-the figure legends are far too brief and very unhelpful-what little text there is simply reiterates the conclusions in the results section. There is no specific statistical information in the legends nor quoting of p values, and little explanation of the experiment.” **We have improved the legend text as requested, working within the journal style limits.**

“-In 1i saying the Glu release is ‘insensitive’ to TBOA is not really correct-more accurately it is not inhibited-TBOA is increasing glutamate levels presumably by preventing re-uptake.

-Is 1j mislabelled (should be bafilomycin not TBOA?)” **We have corrected the point regarding TBOA sensitivity (Figure legend 1). 1j is correct, the second glutamate rise is evoked in the presence of TBOA and bafilomycin.**

Reviewer #4:

In this study, Doyle et al., re-envisage the role of glutamate and glutamate receptors in white matter ischemia, and determine the release mechanism of glutamate in the white matter. The authors elegantly show, using a combination of electrophysiology and imaging, that in ex-vivo glutamate concentrations increase in the white matter during high potassium application (mimicking high neuronal activity) and in ischemia, and blocking NMDA receptors increases action potential propagation recovery from ischemic insults. The authors clearly demonstrate that glutamate is released in the white matter and argue that this is via a vesicular release in myelinated axons into the periaxonal space both during high potassium application and in ischemia; thus, confirming Micu et al., 2016 finding (Micu et al., *Experimental Neurology* 2016; 276, 41–50). The authors further identify a novel NMDA- GluN2C/D subunit blocker that significantly reduces ischemic white matter damage in ex vivo and in vivo, which might be therapeutically significant, thereby reawakening the notion of therapeutic NMDA receptor blockage for therapy after ischemia. Therefore, this study is of considerable interest, particularly due to the therapeutic potential for white matter ischemia. However, regrettably, the study does not sufficiently address the recent papers questioning the role of NMDA receptors and/or glutamate release in white matter ischemia. To further address these papers and to strengthen the conclusions made within the manuscript further evidence is needed, as listed below.

Major comments

(1) The role of glutamate and specifically NMDA receptors in the white matter has been under considerable debate over the last 10 years, and in recent years two major papers cast doubt on previous reports. Saab et al, showed that knocking out NMDA receptors in oligodendrocytes made white matter ischemia outcome worse, not better (Saab et al., 2016 *Neuron*, 91(1), 119–132.) and Hamilton et al., argued that the main damage in white matter ischemia is mediated via TRPA1 channels, not

NMDA receptors and glutamate release (Hamilton et al., 2016 Nature, 529, 523–527). The authors need to better address or discuss, the differences between this report, that is in line with earlier reports, and recently published papers.

Particularly, with regards to TRPA1 channels. In this manuscript blocking TRPA1 channel did not show a functional effect on the recovery of the propagation of the action potential in the RON, nor did it prevent ischemia-induced swelling of myelin, whereas in Hamilton et al., (Nature 2016), blocking TRPA1 channels significantly prevented myelin lamella separation seen after ischemia (see also comment (2)).

The authors did note that a short pre-incubation with NMDA receptor blocker MK801 might explain the difference between previous reports and this manuscript, although this may account for the difference from Baltan et al. (J. Neurosci 2008, 28:1479-1489) and this manuscript, it does not explain the difference of the findings in SAAB et al., (Neuron 2016) which did not use blockers but genetically deleted GluRN1 from oligodendrocytes.

These differences need to be better discussed and addressed.”

We have now extended the discussion to address these highly relevant points made by the reviewer (page 6, line 19). Regarding the Saab paper, conditional ablation of oligodendroglial Nr1 indeed elevated ischemia-sensitivity and lead to eventual white matter injury in its own right, due to chronically impaired oligodendrocyte glucose uptake via GLUT-1. The constitutive Nr1 KO has a major effect upon myelination regulation (Lundgaard et al 2013), and these compensatory events may explain the absence of protection in this model. The paper by Hamilton et al., is unusual in that they find no increase in G-ratio in ischemic optic nerve but report significant TRPA1-insensitive “axon vacuolization”. Looking at the micrographs, this appears to be the same phenomenon we have interpreted as myelin decompaction and bubbling which we now support with ultrastructural evidence (Fig 5 c, d).

“(2) In Figure 3f myelin width is measured, and it is not clear to me as to what is being measured, nor how it is measured (not explained in methods). Given the expertise of the group with electron microscopy (EM) of myelin in the optic nerve, I find it surprising that myelin integrity with EM was not measured, which is a more reliable method and would directly address whether there are differences in myelin lamella separation after ischemia as seen in Hamilton 2016 or not, and whether QNZ-46 improves myelin integrity. Similarly, in figure 5g, low mag staining with Luxol fast blue and level of dye uptake was measured by optical density as a readout of myelin integrity, which does not provide conclusive evidence of myelin integrity, this should be done by electron microscopy and myelin integrity quantified.” **We have now included this EM analysis as requested (Figures 2, 8), and have clarified the methods description of the confocal myelin analysis.**

“(3) Figure 1 a-f, and Suppl.Fig. 1d, show an inconsistency of FM4-64 loading, in Figure 1c and Suppl.Fig. 1d top panel, there is punctate staining, whereas in Figure 1d and Suppl.Fig. 1d lower panel it shows a continuous staining. This continuous staining is interpreted as evidence for that the axonal vesicular release occurs under

the myelin sheaths into periaxonal space – pg2, line 8 “the majority of vesicular fusion must occur under the myelin sheath and empty into the periaxonal space” and on page 3, line 8 “The absence of focal sites of vesicular fusion in axons indicate that ischemic glutamate release empties into the periaxonal space under the myelin sheath.”

Yet most experiments using FM4-64 labelling with 2photon microscopes detect punctate staining, and so did Micu et al., (Experimental Neurology 2016) in the myelinated optic nerve. However, this concern of the specificity of the labelling is mitigated somewhat by the axon-specific reduction in staining and the constant level of FM4-64 staining in astrocytes (although how this was quantified is not clearly explained in the manuscript), and the concomitant measured increase in glutamate within in the optic nerve occurring at a similar timeframe. However, concluding that this continuous type of FM4-64 loading is evidence for release under the myelin and into the periaxonal space is somewhat an over interpretation. But given that over 90% of the axon surface is covered with myelin, particularly in the optic nerve, it is thus likely that the release is under the myelin like Micu et al. argued. Nonetheless, clear evidence for vesicular release into the periaxonal space is not provided thus these statements in the manuscript should be left out, or toned down and only speculated on. Whether it is or not is not essential to the core finding of the manuscript thus the authors should rather focus their efforts to clearly demonstrate that the de-staining of FM4-64 is affected by blockers of vesicular release – this would truly strengthen their core finding that vesicular release from axons triggers myelin damage. **“ We have now included a quantitative analysis of myelin integrity after 30 and 60 min of OGD in the isolated optic nerve preparation (new Fig. 5) and in the contra- and ipsi-lateral external capsule-motor cortex region of the *in vivo* stroke model (new Fig. 8). We have also clarified the analysis of FM4-64 imaging of astrocytes (page 9, line 34). Regarding “punctate” FM4-64 staining in Fig. 1 c and S Fig. 1 d, these have different causes. Fig. 1 c is a low power image to show the general disposition of uptake in a large area and the red channel has consequently pixilated to a degree. When individual L-S axons were focused on, punctate staining was not observed along the axons longitudinal axis. Axons in S Fig. 1 d are mainly caught in X-S, as seen in Fig. 6 and Fig 8 with other imaging techniques, and each “puncta” is in fact a single axon profile. We have now made this point in the text (page 2, line 21).**

“(4) The finding that QNZ-46 reduces stroke volume and improves behavior in mice after 60 min of middle cerebral artery occlusion is the major finding of this paper, and the most exciting. Therefore, it is surprising the lack of detail of the behavior in figure 5, and ultrastructural analysis of myelin integrity (as mentioned above). The manuscript would significantly improve if more detail would be given to the behavioral tests and their outcome, and the effect of QNZ-46 on neurons (or grey matter) to strengthen the comments made in the manuscript regarding using QNZ-46 as a prophylactic therapy approach in patients at risk for stroke.” **In response to the reviewers concern, we have performed an extensive ultrastructure analysis to the *in vivo* stroke model in the presence and absence of QNZ-46 reveals the precise sub-cellular features of the protective effect. By focusing on the external capsule-motor cortex border we are able to describe the nature of the protection the drug provides to neuronal somata, neuropil, glial cell somata, glial processes, myelin and the axons themselves (new Fig. 8).**

“Minor comments

(1) Results section- page2, line 26 “Resting extracellular glutamate was 1.6 +/- 1.7 mM in adult CC, and TBOA evoked a 1.3 +/-0.4 mM peak increase..” is confusing. When comparing this data to the data presented in supplementary figure 2, it is not clear as to whether the value 1.6 +/-1.7 mM of resting extracellular glutamate relates to Suppl.Fig. 2a, which in fact shows around 1.6mM of resting glutamate levels, or whether it should relate to suppl. Fig. 2b, where control max change in [glutamate] in ‘resting’ condition before TBOA are given as ~0.5mM, as the max change in glutamate levels in TBOA in that figure is ~1.3mM – same as in the text. Are the authors comparing ‘resting’ [glutamate] levels in Suppl.Fig. 2a to TBOA max change in [glutamate] levels in Suppl.Fig. 2b. – This is confusing and needs better clarification in the text as supplementary figures 2a and 2b are not showing the same (2a is showing measured levels in [glutamate], and 2b is showing the max change in [glutamate]).” **We have now changed the text to improve clarity and we apologize for the confusion in the original (page 2, line 25).**

“(2) In Figure 2a, the trace is missing for recordings in solutions lacking Na+, and in 2b the trace recorded with Diltiazem is missing.” **The traces for zero Na+ and diltiazem have been omitted for clarity with the means shown on the histogram.**

“(3) Figure 2- it would be clearer if the bar graph for Rose Bengal and bafilomycin/tboa was labelled as (d), as it presumably it is figure 2d! “ **We now corrected this figure label as requested.**

“(4) Page 3 line 32 - “Myelin sheath diameter increased from 0.67 mm +/-0.02 to 0.90 mm +/-0.07 after 60 min ischemia + 60 min recovery, an effect that was prevented by 120 min pre-treatment with QNZ-46 but not by TRP1A block (Fig. 3 e, f)” – the unit should be in μm not mm. “ **We have corrected this typo.**

“(5) There are a number of typos and errors within the text and figures, such as supplementary figure 2a, where RON should be MON, when showing the glutamate concentration for mouse optic nerve, instead of rat optic nerve. - and on page 3, line 3 NPPD should be NPPB, etc..” **We have corrected these errors and have carefully re-read the text. We thank the reviewer for their attention to detail.**

Yours,

Robert Fern.
Professor of Translational Neurobiology
Peninsular School of Medicine and Dentistry
University of Plymouth

Reviewers' comments:

Reviewer #1 (Remarks to the Author):

In this revised manuscript, the authors have added a significant amount of new data in response to the previous comments of the reviewers. The study provides strong evidence that glutamate release occurs through vesicular release along axons and results in elevated glutamate levels in the peri-axonal space. The authors propose that a major target of axonal vesicular glutamate is the myelin sheath that rapidly becomes disrupted and decompacted resulting in further functional deficits following excessive glutamate release. Pre-treatment with QNZ-46 that targets GluN2C/D NMDA receptors largely inhibited these pathological changes suggesting that such related therapies may be effective under excitotoxicity conditions such as stroke.

While the authors have put considerable work into the revised manuscript there remains a number of concerns. The major concern relates to the demonstration of myelin as the primary target of glutamate damage. The authors provide EM studies from the rat optic nerve to support the notion of vesicle release (Fig 2). Unfortunately, these are not particularly convincing. Further and more importantly the changes shown in Fig 5 are also not compelling. The pathology in 5c looks to be more axonal (shrunken axons) than myelin and the evidence of wide-spread myelin damage is still missing. Indeed, the data in Fig. 8 is more consistent with neuronal/axonal protection than with myelin protection. While it is extremely difficult to unambiguously identify primary pathological targets in the models used by the authors, without clear evidence of a direct effect on myelin and significant disruption the study remains speculative.

Reviewer #2 (Remarks to the Author):

The authors have added extensive new data to support their conclusions. Of special interest is new EM data (Figure 2) showing vesicles clustered in the sub-myelinic axoplasm, which are apparently reduced in number following transient oxygen glucose deprivation. While the quantitative data set is small and should probably be viewed as preliminary, these observations support the overall hypothesis.

The authors have satisfactorily addressed comments in Review 2, largely through clarifications in the text.

The manuscript demonstrates significant stroke neuroprotection with QNZ-46 pretreatment. However it still seems speculative to consider chronic NMDA receptor blockade (of any receptor subtype) as a clinically tolerated treatment for stroke prophylaxis. The discussion is an appropriate place for such speculation but I think the final sentence of the abstract overstates the case.

Reviewer #3 (Remarks to the Author):

The authors have made significant efforts to address the referees' issues and are to be applauded for this. The m/s is considerably stronger.

My main remaining concern is around the claim that the C/D antagonist may represent a "low-impact prophylactic therapy" as stated in the abstract and elsewhere.

The authors provide no evidence that prolonged treatment of this drug is 'low-impact'. Indeed the work of the Nave lab (e.g Saab 2016, Neuron) shows the very harmful consequences on functional integrity and outcome after ischemia when NMDAR activity is chronically interfered with.

Reviewer #4 (Remarks to the Author):

The manuscript has improved somewhat, but regrettably, my comments have not been sufficiently addressed.

The manuscript would still benefit from further clarity and focus. Most importantly the findings

should be better linked to the existing literature. The findings in Figures 1 and 2a,b are not novel and should be presented as such and relevant literature cited (as per by my previous review). The novelty is that glutamate release in ischemia may be via vesicular release and the prevention of QNZ-46 against ischemia mediated white matter (and grey matter) damage (seen in figure 4,6-8). This should be better discussed, and linked to existing literature as this manuscript presents a series of data contradicting previous findings. The findings in this manuscript need to be better presented and discussed.

Reviewer #1 (Remarks to the Author):

In this revised manuscript, the authors have added a significant amount of new data in response to the previous comments of the reviewers. The study provides strong evidence that glutamate release occurs through vesicular release along axons and results in elevated glutamate levels in the peri-axonal space. The authors propose that a major target of axonal vesicular glutamate is the myelin sheath that rapidly becomes disrupted and decompacted resulting in further functional deficits following excessive glutamate release. Pre-treatment with QNZ-46 that targets GluN2C/D NMDA receptors largely inhibited these pathological changes suggesting that such related therapies may be effective under excitotoxicity conditions such as stroke.

While the authors have put considerable work into the revised manuscript there remains a number of concerns. The major concern relates to the demonstration of myelin as the primary target of glutamate damage. The authors provide EM studies from the rat optic nerve to support the notion of vesicle release (Fig 2). Unfortunately, these are not particularly convincing. Further and more importantly the changes shown in Fig 5 are also not compelling. The pathology in 5c looks to be more axonal (shrunken axons) than myelin and the evidence of wide-spread myelin damage is still missing. Indeed, the data in Fig. 8 is more consistent with neuronal/axonal protection than with myelin protection. While it is extremely difficult to unambiguously identify primary pathological targets in the models used by the authors, without clear evidence of a direct effect on myelin and significant disruption the study remains speculative.

We have addressed the reviewer's concern that the primary target of glutamate damage should be shown to be myelin. We have now extended the ultrastructural analysis in Figure 5 to demonstrate that there is no significant diameter reduction or shrinkage of the axon cylinder following OGD (new Fig. 5 k). This demonstrates that the analysis in panels g-j where g-ratio is affected across the axon diameter spectrum is definitive proof of wide-spread myelin pathology; and that this is prevented by QNZ-46. We have also included new data (new Fig 5 l) showing low-power myelin staining that is shown to be significantly reduced following OGD; this effect is also prevented by GluR block.

In addition, we would like to point out that we have quantified loss of myelin following MCAO in Fig. 7 g-j while we have now included a new supplementary Figure (new Figure S 7) to provide further ultrastructural evidence for wide-scale myelin damage following MCAO in this model (and its prevention by QNZ-46). We feel that these changes provide multiple and persuasive evidence for wide-scale glutamate-mediated myelin damage following OGD as requested by the reviewer.

Reviewer #2 (Remarks to the Author):

The authors have added extensive new data to support their conclusions. Of special interest is new EM data (Figure 2) showing vesicles clustered in the sub-myelinic axoplasm, which are apparently reduced in number following transient oxygen glucose deprivation. While the quantitative data set is small and should probably be viewed as preliminary, these observations support the overall hypothesis.

The authors have satisfactorily addressed comments in Review 2, largely through clarifications in the text.

The manuscript demonstrates significant stroke neuroprotection with QNZ-46 pretreatment. However it still seems speculative to consider chronic NMDA receptor blockade (of any receptor subtype) as a clinically tolerated treatment for stroke prophylaxis. The discussion is an appropriate place for such speculation but I think the final sentence of the abstract overstates the case.

We have now modulated the final sentence of the abstract in line with the reviewer's suggestion.

Reviewer #3 (Remarks to the Author):

The authors have made significant efforts to address the referees' issues and are to be applauded for this. The m/s is considerably stronger.

My main remaining concern is around the claim that the C/D antagonist may represent a "low-impact prophylactic therapy" as stated in the abstract and elsewhere.

The authors provide no evidence that prolonged treatment of this drug is 'low-impact'. Indeed the work of the Nave lab (e.g Saab 2016, Neuron) shows the very harmful consequences on functional integrity and outcome after ischemia when NMDAR activity is chronically interfered with.

We have now modulated the final sentence of the abstract in line with the reviewer's suggestion.

Reviewer #4 (Remarks to the Author):

The manuscript has improved somewhat, but regrettably, my comments have not been sufficiently addressed.

The manuscript would still benefit from further clarity and focus. Most importantly the findings should be better linked to the existing literature. The findings in Figures 1 and 2a,b are not novel and should be presented as such and relevant literature cited (as per by my previous review). The novelty is that glutamate release in ischemia may be via vesicular release and the prevention of QNZ-46 against ischemia mediated white matter (and grey matter) damage (seen in figure 4,6-8). This should be better discussed, and linked to existing literature as this manuscript presents a series of data contradicting previous findings. The findings in this manuscript need to be better presented and discussed.

The reviewer is concerned that the data in Figure 1 and 2 a, b are not novel, in that axonal release of vesicular glutamate has been documented previously (Kukley et al., 2007) as has NMDA receptor mediated calcium elevation in myelin (Micu et al., 2006). The data in Figure 1 and 2, however, are the first to document vesicular fusion in axons in adult brain and to record the glutamate rise associated with this event. They are also essential preliminary

steps in testing the hypothesis that these events are central to myelin pathology. The reviewer also requires better citation of the relevant work and we have improved these aspects in various places through the text, including a new short paragraph in the discussion.

We feel that these changes have significantly strengthened the manuscript and respectfully re-submit it for publication in Nature Communications.

REVIEWERS' COMMENTS:

Reviewer #1 (Remarks to the Author):

In response to the latest reviews of the manuscript the authors have added additional data to demonstrate a selective change in myelin ultrastructure following OGD and interpret this as evidence that myelin is a direct target of elevated glutamate levels. Such data is clearly consistent with the interpretations of the authors and provides additional support for the hypothesis. The authors have also added additional citations and modified the text in several places. Such changes have improved the manuscript.

Reviewer #4 (Remarks to the Author):

I still have concerns over some of the quality of some the data presented as per the first review. However, the electrophysiological, and the glutamate/ion sensing data, and the data presented in figure 7 are solid and give support to some of the conclusions and some of the added data has improved the manuscript.

In addition, as per my first review, the authors should highlight the similarities of their findings to that of Micu et al., *Experimental Neurology* 2016.

Response to Reviewer 4 comment:

“I still have concerns over some of the quality of some the data presented as per the first review. However, the electrophysiological, and the glutamate/ion sensing data, and the data presented in figure 7 are solid and give support to some of the conclusions and some of the added data has improved the manuscript.

In addition, as per my first review, the authors should highlight the similarities of their findings to that of Micu et al., *Experimental Neurology* 2016.”

We have now added the citation in an appropriate context as requested by reviewer 4 (new citation 5).